# The Utilization of Jackfruit (*Artocarpus heterophyllus* L.) Waste towards Sustainable Energy and Biochemicals: The Attainment of Zero-Waste Technologies

**Prakash Kumar Sarangi** [1,†] , **Rajesh Kumar Srivastava** [2,†] , **Akhilesh Kumar Singh** [3,*] ,
**Uttam Kumar Sahoo** [4,*] , **Piotr Prus** [5,*] and **Paweł Dziekański** [6]

1   College of Agriculture, Central Agricultural University, Imphal 795004, India; sarangi77@yahoo.co.in
2   Department of Biotechnology, Gandhi Institute of Technology and Management (GITAM),
    Visakhapatnam 530045, India; rajeshksrivastava73@yahoo.co.in
3   Department of Biotechnology, Mahatma Gandhi Central University, Motihari 845401, India
4   Department of Forestry, Mizoram University, Aizawl 796004, India
5   Department of Agronomy, Faculty of Agriculture and Biotechnology, Bydgoszcz University of Science and
    Technology, Al. Prof. S. Kaliskiego 7, 85-796 Bydgoszcz, Poland
6   Department of Economics and Finance, Jan Kochanowski University in Kielce, 25-369 Kielce, Poland;
    pawel.dziekanski@ujk.edu.pl
*   Correspondence: akhiliit@gmail.com (A.K.S.); uttams64@gmail.com (U.K.S.); piotr.prus@pbs.edu.pl (P.P.)
†   These authors contributed equally to this work.

**Abstract:** The valorisation of food and fruit wastes has the potential for the production of sustainable energy and biochemicals. Approximately 70% of the weight of the original jackfruit (*Artocarpus heterophyllus* L.) fruit is lost during its processing as waste in the form of peeled skin and core, both of which have not been utilized and, thus these contribute to disposal as well as pollution issues. The major components such as cellulose and hemicellulose can be easily biologically transformed into bioenergy sources such as ethanol, methanol, and butanol; valuable phenolics and biotechnological products such as pectin, citric acid, bromelain, ferulic acid, and vanillin; and many other products. These residues can also be utilized as essential sources for the biological transformation process, leading to the production of numerous products with added value, such as phenolic antioxidants, phenolic flavour compounds, and organic acids. Thus, the value addition of jackfruit waste can support sustainable solutions towards food and nutritional security. In this way, zero waste can be achieved through novel biorefineries, which are critically highlighted in this paper. Furthermore, novel technologies for the conversion of jackfruit waste are summarized with recent findings.

**Keywords:** jackfruit waste; zero waste; bioconversion; bioenergy; phenolic; biochemical

## 1. Introduction

Jackfruit (*Artocarpus heterophyllus* L.) trees are known to produce huge fruits from their stems and are unique in terms of food utilization as vegetables and fruits. Jackfruit trees are cultivated throughout the lowlands in south and southeast Asian countries. In addition, it is found in east Africa, as well as throughout Brazil and Caribbean nations such as Jamaica. Many countries such as Malayasia, the Philippines, Cambodia, and Sri Lanka, including Bangladesh (where jackfruit is the national fruit), are taking aggressive efforts to bring about greater areas for jackfruit plantation; therefore, the production and productivity of jackfruit are likely to increase over the years, as will jackfruit waste. Jackfruit is one of the most demanded fruit crops in India too. In India, it is grown/cultivated in low-elevation regions throughout all the states [1]. Several wastes are generated from its ripe fruits and this fruit waste is more palatable than waste from raw fruit [1,2]. Jackfruit has shown various nutrients such as crude protein (CP ~7.9%), crude fibre (CF ~14.1%), calcium (0.8%), and phosphorus (0.1%). Except for these nutrients, ripe and raw jackfruit wastes

are also utilized as potential substrates for energy production and nitrogen-free extracts (NFE ~65%). The rind of the ripe fruit is a good source for cattle foods and this waste of jackfruit is a non-edible portion (~59.2%), such as perianth meal, rind, and core meal. This waste matter is utilized for total dry meal recovery (11.6%) [2]. Some analyses have been performed on jackfruits waste compositions (for perianth meal, rind, and core meal), which contain ash (6 to 7.5%), carbohydrates (20–29%), crude protein (8–10.6%), crude fate (1.7 to 7.3%), and crude fibres (12–17.3%). Some studies have been conducted on the utilization of jackfruit wastes and these are used for food, feed, and other industry applications. Some non-edible portions of jackfruit (such as peels and axis) are reported with edible products (seed), and still, these waste matters are underutilized worldwide such as in Bangladesh, India, and other countries [3]. Different waste portions of jackfruits are generated during juicy edible bulbs. The thick peels of jackfruits can be utilized for different valuable product generations such as biofuel, non-porous adsorbents, and nutrient-enriched cattle feeds. Normally, non-porous adsorbent is used in the removal of dye. The peel and central axis of the jackfruit are also utilized for pectin extraction [4]. Its seed power is reported to be used in various bakery products. From this fruit, seed powder, starch, and protein fraction are isolated and then utilized in their purified form for food formulations at the industry level [1,3,4].

While jackfruit has massive potential with unlimited benefits, its waste products are much more than those of any other tropical fruits. According to Sundarraj and Ranganathan [5], 75% of the jackfruit products in India get wasted due to inadequate marketing, negligence, and a lack of processing facilities, and it is estimated that almost 20,000 million Indian rupees worth of jackfruit waste are reported only in the two Indian states of Karnataka and Kerala, which are major jackfruit-producing states. The non-edible parts of jackfruit that form the waste, however, can be made effective to use by bioconversion using various technologies. For example, the thick peel of jackfruit can be converted into nutrient-enriched cattle feed, for the extraction of biofuel, or into a nano-porous absorbent for removing dye, etc. The fruit waste can be a promising source of food and feed. Though several reports are available on value-added products from the edible parts of jackfruit, the conversion of jackfruit waste into various new value-added products using the latest technological interventions is somewhat limited. Thus, there is an urgent need to document and make a comprehensive review on the enormous jackfruit waste during the pre- and post-harvest stages and the use of this waste towards sustainable energy generation, leading to zero waste via various novel technologies.

## 2. Literature Review

Jackfruits have been analysed during different seasons by various workers [4–8] for their nutritional and antioxidant properties. The findings of these studies have indicated that jackfruits serve as a valuable source in the development of nutraceuticals, which are currently in high demand worldwide [6]. Jackfruits in different seasons are reported to differ in phytochemicals such as phenolics, terpenoids, steroids, glycosides, saponins, alkaloids, and tannins and these compounds are known to possess antioxidant properties. Diversity in the secondary metabolites in jackfruit has been reported and attributed to variations in functionally, nutritionally, and medically important jackfruit wastes [4,6]. From jackfruit waste, antibacterial and antioxidant activity agents/compounds can be evaluated from the extracts that are obtained from methanol extract from jackfruit leaves and stem barks, and then they are applied as a peel-off mask. From the extraction of these properties from jackfruit waste, the first raw material was macerated using the methanol agent and then filtrates were evaporated for some time to obtain a concentrated crude extract [6]. This extract was evaluated using different tests such as phytochemistry screening and also antibacterial tests on *Propionibakerium acnes* and *Staphylococcus aureus* at different concentrations of extracts using the DPPH (a,a-diphenyl-β-picrylhydrazyl) method [8].

For the best evaluation of these phytochemicals, some additional tests have been performed that can prove the characteristics of peel-off masks, such as homogeneity, pH, organoleptic, and irritation tests. Some phytochemical screenings have proved the domination of tannin and saponin in these extracts [7,8]. In the context of jackfruit waste, these are rich sources of carbohydrates, protein, fats, and phytochemicals. This organic extraction or utilization serves as a promising feedstock for valuable bioproducts, including fuels/chemicals synthesis. Several pre-treatments (such as biological, physical, and chemical, including green solvents) have been applied as effective valorisation strategies for jackfruit waste matters [9]. The implementation of these strategies has facilitated the transformation of waste into products that possess added value, including, but not limited, to bioethanol, biogas, bioplastic, feeds, and functional compounds/food additives. The utilization of jackfruit waste for bioenergy production and recovery represents a promising avenue for sustainable and eco-friendly food waste-based renewable resources. This approach offers an economically feasible alternative to non-renewable fossil fuels [10]. Further efforts have been performed on efficient bioconversion tasks/techniques, applied for jackfruits that can generate/produce valuable biomaterials/chemicals, and this is only to be achieved via a microbial fermentation process. This conversion can help to obtain sustainable products with the mitigation of jackfruit generation/accumulation and support for a green environment. Some reports have claimed the utility of jackfruit peel for the remediation of dye colour from contaminated aquatic environments [8,9]. The implementation of said technology has the potential to facilitate the creation of an environmentally sustainable economic framework cantered on the repurposing of waste materials. Many studies have been carried out on the utilization of jackfruit waste for the production of value-added products, with the ultimate goal of mitigating waste generation and promoting environmental sustainability. In an attempt to utilize jackfruit waste, the production of plastic from jackfruit seed starch has also been carried out. The resulting plastic was strengthened through the incorporation of microcrystalline cellulose (MCC) derived from cocoa pod husks, with glycerol serving as the plasticizer [11]. This study aimed to identify the optimal mass and volume of microcrystalline cellulose (MCC) and glycerol concentration for the production of bioplastics in a high yield. Before the bioplastic production, MCC bio-production was achieved with cocoa pod husks, and these husks were subjected to a pre-treatment task with the help of alkali agents, bleaching, and an HCl acid solution to obtain effective hydrolysis [12]. In this, the degree of crystallinity of the MCC determination was carried out with the help of analytical techniques such as XRD (X-ray diffraction), with a functional groups determination using FTIR (Fourier-transform infrared) and also a morphological properties analysis using scanning electron microscopy (SEM). Some researchers have relied on results for isolated MCC from pod husks and discussed it as a rod-like form with a respective length (5–10 μm) and diameter (11.63 nm) with a high crystallinity [11,13]. From the isolated MCC utility in bioplastic synthesis, the tensile property of bioplastic was determined at a starch to MCC mass ratio (8:2). Further tasks were performed in addition to 20% glycerol with a measured tensile strength (0.637) and good elongation (at break of 7.04%) [14]. From analytical measurements using FTIR spectroscopy for bioplastic functional groups studies, greater numbers of -OH groups were found in bioplastics, and these were reinforced with filler MCC, with the representation of a hydrogen bond [11,14].

Biofuels such as biodiesel, an eco-friendly and renewable biofuel, have emerged as a cutting-edge substitute for petroleum-based diesel. This fuel possesses comparable traits to conventional fossil fuels and exhibits a remarkably low emission profile. The adoption of biodiesel not only seeks to diminish the reliance on non-renewable resources, but also fosters economic growth while boosting energy security, as detailed previously [15]. Nevertheless, through an exhaustive exploration of the existing scientific literature, it becomes evident that the untapped potential of jackfruit waste in promoting sustainable energy and achieving zero waste remains largely unexplored [16]. The scarcity of research has left unanswered questions about the true potential of jackfruit waste as a feedstock for combustion and bioenergy generation [16]. To bridge this knowledge gap, it becomes im-

perative to delve into the scientific exploration of jackfruit waste combustion. For instance, several research endeavours have investigated the untapped possibilities of jackfruit and its byproducts, encompassing edible fruit, seeds, peels, and latex-like filaments (rags or perianth) [17,18]. These comprehensive investigations have been centred on evaluating the nutritional content, mineral composition, and physicochemical properties of these diverse components. However, previous studies have convincingly established that jackfruit and its waste harbour essential elements such as potassium, magnesium, and calcium, which play pivotal roles in facilitating the catalytic process of biodiesel synthesis [15,17]. In the realm of scientific research, an intriguing area yet to be fully explored lies in the use of jackfruit waste, such as jackfruit peel waste (JPW), as a catalyst in biodiesel production. Presently, the open literature is conspicuously void of significant investigations into the production and application of JPW biomass catalysts, creating a stimulating challenge for researchers [16,17]. The scarcity of information concerning JPW biomass catalysts serves as the impetus behind investigations on the same, as it endeavours to bridge the existing knowledge gap in the realm of catalyst development for biodiesel production. By focusing on JPW, an abundantly available agricultural waste material, the research attempts to transform it into a valuable resource, i.e., catalysts that proficiently convert waste cooking oil (WCO) into biodiesel [16–18]. This is supported by the fact that using the $K_2O$ from JPW ash as a cost-effective solid catalyst for biodiesel generation is feasible. By leveraging JPW biomass as a distinct solid catalyst, this approach bears several merits. Foremost, it upholds environmental friendliness by repurposing an agricultural waste product that would otherwise be discarded, thus contributing to waste reduction and fostering sustainability in biodiesel production [18]. The catalyst proves renewable, reusable, recyclable, non-hazardous, and environmentally benign, with vast applications across diverse fields. Aligned with these principles, Mulkan et al. [15] diligently conducted their investigation, culminating in compelling evidence showcasing the successful application of JPW as a solid catalyst for biodiesel synthesis. Further, the United Nations has taken significant steps toward ensuring a sustainable future for the entire world by 2030 [15,19]. One of its key initiatives involves the establishment of the Sustainable Development Goals (SDGs), with SDG 7 specifically focusing on promoting the sustainable utilization of bioresources to increase the proportion of renewable energy in the global energy mix. Additionally, SDG 7 aims to provide sustainable energy services to all nations [15,18]. Over the past few decades, considerable efforts have been made to adopt cleaner and greener technologies that harness various bioresources. These endeavours have led to the continuous implementation of innovative technologies [18,19]. Considering these, the present review aimed to highlight the approaches/methods (with their advantages, limitations, and drawbacks) employed for the transformation of jackfruit waste into sustainable energy/biochemicals and advances made in the field, together with the emerging trend of utilizing jackfruit waste as a bio-absorbent to combat and alleviate pollution-related issues.

## 3. Methodology and Analysis

In this comprehensive scientific review, an extensive search across various renowned databases was conducted, including the ISI Web of Science and Google Scholar, to gather a wealth of published articles on diverse aspects of jackfruit cultivation and utilization. In this context, the focus encompassed explorations of the nutritional properties of jackfruit seeds, the composition of jackfruit waste, and the different types of wastes generated [20]. Furthermore, various valorisation strategies were searched, involving physical, biological, and chemical techniques together with innovative green solvents. These cutting-edge techniques serve as vital conduits in extracting valuable bioactive compounds, such as phenolics, carotenoids, flavonoids, tannins, antioxidants, and more, from the discarded remnants of the fruit [14,16]. Apart from this, discussions are being made about the promising potential of microbial fermentation for transforming jackfruit waste into valuable resources. Such a transformative process unlocks the possibility of harnessing bioethanol, biogas, bioplastics, and other value-added products, thus paving the way for a sustainable and eco-friendly

approach to waste management [21]. Our scrutiny extended to exploring the practical implementations of conversion technologies for jackfruit waste, meticulously evaluating their cost-effectiveness and limitations within the broader global context. Throughout this review, we elucidated how the utilization of jackfruit waste holds immense promise for augmenting the share of renewable energy in the global energy mix while ensuring a strict adherence to environmental safeguards [19,20]. Altogether, this review has provided a compelling description that not only highlighted the abundant potential of jackfruit waste valorisation, but also emphasized its pivotal role in fostering a greener, more sustainable future for our planet [20,21].

## 4. Results and Discussion

### 4.1. Cultivation of Jackfruits with Impact on Its Nutrients

Due to more demand for waste mitigation and also the extraction of valuable products with bioactive properties, there is more jackfruit cultivation being reported all over India and also in some other countries, such as Myanmar, Malaysia, and some locations in Brazil. It is now one of the most remunerative and also more valuable fruits in India. The jackfruit tree belongs to the Moraceae family and is also a native tree of India [22]. This plant is now cultivated throughout the tropical lowlands in both hemispheres. The jackfruit plant is widely grown in the Western Ghats of India, but its plantation is found in Bihar, West Bengal, Uttar Pradesh, Kerala, Tamil Nadu, Assam, and Orissa. Some reports have claimed its regular plantation in the U.P., especially in marginal orchards. However, in other parts, this plant cultivation has been reported as rare in plantations, but jackfruit cultivation is found throughout south India up to an elevation of 2400 m [23]. Some reports have discussed jackfruit plantation in systematic and proper ways, and this plant growth needs rich and well-drained sandy loam soils. For this plant plantation, soil drainage has shown more importance with proper evidence. There has been a sudden decline in numerous jackfruit plants in areas that are suffering from a sudden rise in water level [22,23]. Reports on jackfruit tolerance capacity, especially for moisture stress, have been conducted, and tolerance can be shown, to some extent, to be due to the presence of lime and chlorine areas near river beds and can be found to be ideally suitable for jackfruit plant cultivation. Further information on jackfruit plantation is that it requires a warm, humid plain and is able to flourish on humid hill slopes up to an elevation of 1500 m. Jackfruit can deteriorate at higher altitudes with satisfactory plantations in the arid and warm plains of south India [24].

Jackfruit can provide an unlimited scope for the clone selection of promising strains/species for multiple applications. Many types of these plants are available under various local names and they can be originated via clone selection. Gulabi, Champa, and Hazarix are some examples of jackfruits. Storage conditions are needed for jackfruit plantation. The impact of thick peels is found with a good storage quality [25]. Jackfruits can be easily used for cross-pollination and then their seeds can be found to easily propagate. To date, the prevailing population of jackfruits is distributed across numerous trees, exhibiting variations in their morphological characteristics such as shape, size, and density of tubercles, as well as differences in rind colour, bulb size, fibre content, and fruit quality and maturity [23,25]. The storage longevity of jackfruit has been determined to be approximately 6 weeks when exposed to temperatures ranging from 0.1 to 12.7 °C, provided that the humidity level is maintained at eighty to ninety percent. The initial quality and maturity stage at the time of harvest are crucial factors that can significantly impact the storage life of a product. Jackfruit's storage life can be extended, allowing for its transportation to remote locations for marketing purposes, through the utilization of the appropriate packaging and wrapping techniques [24,25].

Reports on jackfruit application in India have shown that they are used as culinary and also table fruits. Jackfruit used for culinary purposes is mainly found in more states in India. In terms of market demand, tender fruits in spring and also summer are used as population vegetables. People have enjoyed jackfruits with a high demand and premier

cost/price, and sometimes this jackfruit cost/price can be reached at a high rate in the year [26]. Most people in the world use jackfruits in their ripe form as tasty fruits with a high sweetness and also high nutritive values. In their ripe form, jackfruits are good sources of vitamins A and C, with some people believing they aid in the digestion process and using them for ailments on a regular basis [25,26].

### 4.2. Compositions of Jackfruits Waste

The waste produced from mature jackfruit fruits has been found to possess a greater palatability compared to the waste generated from its raw fruits. This waste material is composed of crude fibre, crude protein, and various minerals. It is also a significant source of energy, with a substantial quantity of NFE. The generation of rind from ripe fruit can serve as a source of nourishment for cattle [27]. Additional applications of jackfruit-derived fruits have been documented for the production of pickles, dehydrated leather, and thin papad, as well as for canning purposes. These fruits have also been utilized in conjunction with soft drinks such as nectar and squash. In certain studies, rind has been identified as a viable source of protein. Additionally, it has been reported that extracts derived from rind waste can be utilized for the fabrication of jelly [28]. A discussion of the skin parts of jackfruits has reported them as being excellent for cattle feed. Jackfruit plant/timber can be used for making furniture that can exhibit less chance of white ant infection/attacks. Some efforts have been made towards latex extraction that contains resin. The latex from jackfruit bark can be used for making plug holes in earthen vats and baskets and it has shown multiple applications for humankind [27,28].

Further, multiple reports have discussed that jackfruit parts (in terms of their edible and non-edible parts) and non-edible parts are 70–80%, and from this, the outer rind, perianth, and central core of jackfruits are 60% under waste matters. A number of studies have been performed on their biochemical composition using suitable analyses on jackfruit-based wastes with promising sources of health benefits. These bioactive compounds (i.e., valuable compounds/bioproducts) can be recovered/generated using several microbial bioprocessing/extraction techniques in an eco-friendly way [29]. The peel of jackfruit is a rich source of cellulose, protein, starch, and pectin. The chemical composition of dried jackfruit seeds includes carbohydrates (76%), protein (18%), and lipids (2.1%). Numerous phytochemicals, isoflavones, lignin, and saponins, along with several essential nutrients, have been documented in the seeds of jackfruit waste [30]. Subsequent discourse has revealed noteworthy sources of vitamins such as thiamine and riboflavin. The processing of jackfruit generates solid waste, including jackfruit peels and seeds, as well as latex. These waste products have been identified as contributors to environmental damage [30,31]. Researchers have put some effort into jackfruit wastes and they can be used as eco-friendly feedstock/sources for bioproduct syntheses in sustainable ways. This waste matter has shown the best biochemical compositions, helping with renewable modes of bio-based product development. Further research has been conducted on the utilization of jackfruit peels as a sustainable source for the recovery of commercial pectin, biofuels, and other valuable products [32,33]. Figure 1 shows several forms of jackfruit waste with their nutrients.

Various Types of Jackfruits Waste

Waste generation from various types of vegetables and fruits/cultivation processes are discussed, in which a fruit can usually generate from 3.5 to 10 kg of waste in terms of weight, and some fruits can generate higher quantities of waste (up to 25 kg). In the case of jackfruit plants, the outer parts of the jackfruit shell are found to contain a conical apex, covered via a thick, rubbery wall. Under matured conditions, its fruits show a non-edible core that can be found on a longitudinal axis, fused with rags [34]. This part is then fused to the fruit's rind part. Next, the other waste of jackfruit is its bulbs, and these bulbs are composed of pulp, surrounded by seeds and found between the rags. The seed numbers in jackfruits are from nearly 100 to 500 and can be found to be nearly 18 to 25% of the weight

of jackfruits. The kernel of the seeds can constitute from nearly 90 to 95% of their weight and then pulp can account for 30%, and can be found in between 70 and 80% of jackfruits components as non-edible [35]. In regard to the non-edible parts of these fruits, their outer rinds, perianth, and central core can consist of 60% of the total waste of jackfruits as discarded parts. Jackfruit peels and its conical carpel apices are also important components of jackfruit waste. Apart from this waste, jackfruit leaves can be also used for some medical benefits, such as relieving fever, boils, wounds, and skin conditions [34,35]. Then, the young fruits of jackfruit can exhibit acrid and astringent properties and characteristics, and it can help in addressing flatulence in human health. Jackfruit seeds can be found as rich sources of protein concentrations (5 to 6%). In each jackfruit tree, nearly 100 to 200 fruits can be produced, as it is a large and evergreen tree every year. Up until now, very limited research has been performed on the unutilized waste of jackfruits, such as peels and fibre that contribute nearly 60% of the whole fruit, as the largest known edible fruits from any plants. Jackfruit pulp and seeds are known to contain various bioactive compounds in addition to waste nutrients. The availability of these compounds in the fruit of this plant can vary, as reported in a previous study [36]. The cultivation of this plant species is feasible during the monsoon season in coastal areas. The fruit of this plant is considered to be a cost-effective source of sustenance and is widely accessible at a low market value. Subsequent investigations have been conducted on the underutilized segments of jackfruit, with a focus on mitigating the accumulation of biowaste to promote the eradication of pathogenic microorganisms in augmented agricultural yields [37]. The nutritional and functional characteristics of jackfruit peels warrant further investigation for the purpose of extracting bioactive compounds that may be utilized in the pharmaceutical industry [37,38]. Figure 2 shows the different jackfruit wastes with their valorised products.

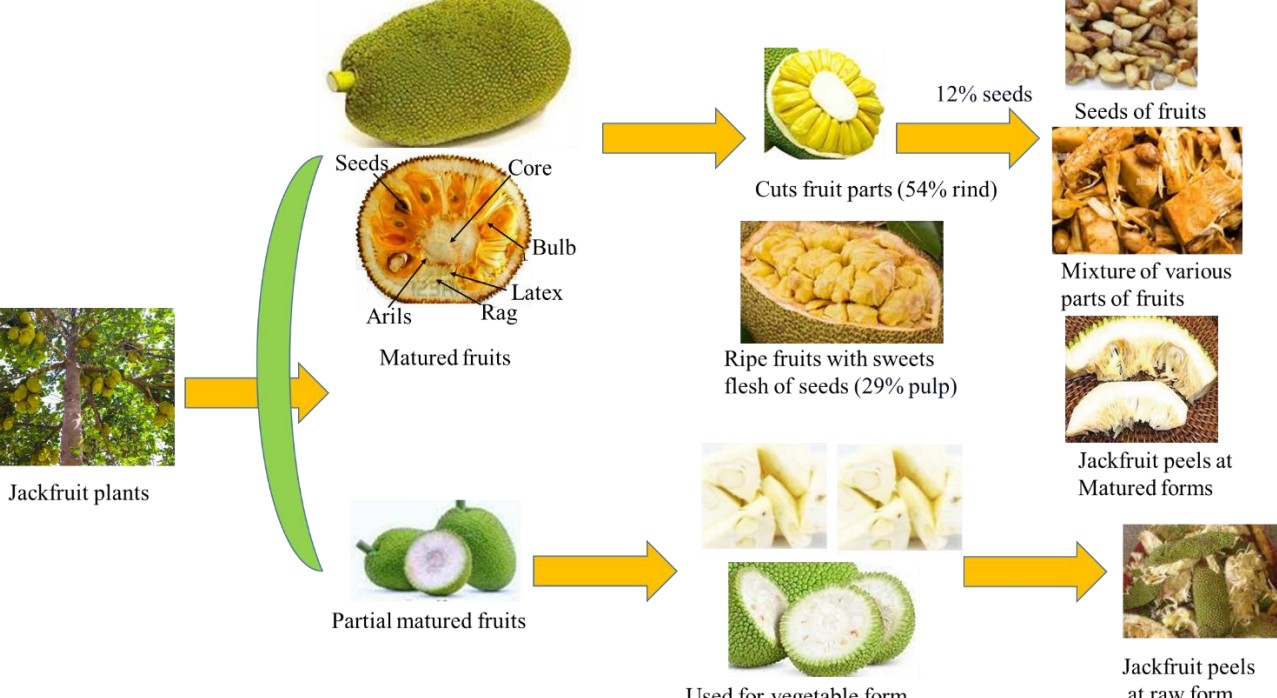

**Figure 1.** Production of jackfruit waste.

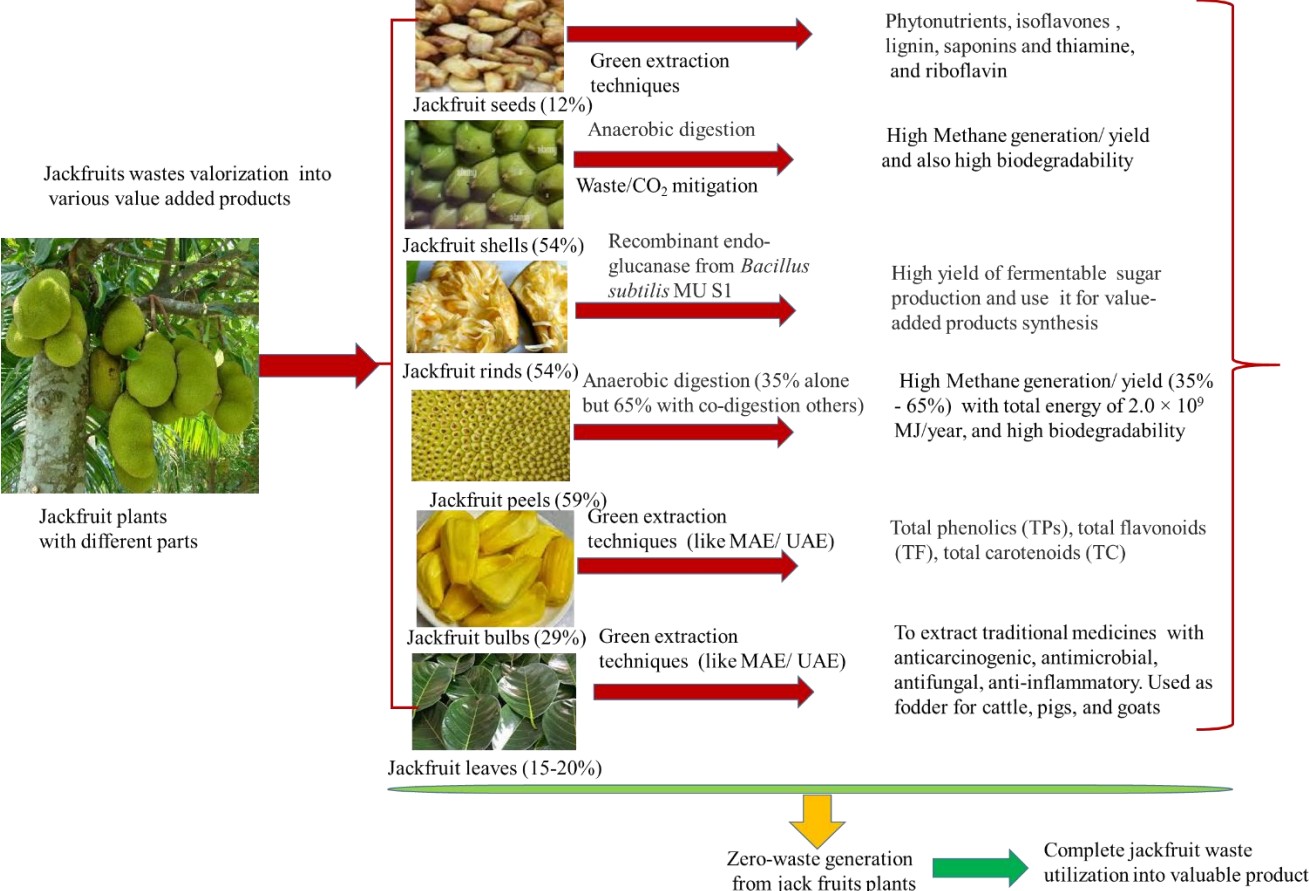

**Figure 2.** Jackfruit waste valorisation into different bioproducts.

The outer layer of jackfruit comprises peel that exhibits a spiky pattern on its surface. The peel of jackfruit is considered to be non-edible and is regarded as a potential waste material. Normally, this part of the fruit is discarded, but recently, it has been used for fertilizer sources. These parts of jackfruit can be used to feed the cattle in villages. It is a good source of carbohydrates (up to 24%), protein (up to 9%), and fibre (17.3%) [39]. Jackfruit peels are utilized as valuable raw materials for functional ingredients such as steroids, triterpenoids, saponins, and carbohydrates. In jackfruit peel, more quantities of polyphenols are reported and this is linked to the peel proximity with extrinsic domains. Some abiotic stress parameters/factors (such as daylight, ultraviolet radiation, and climatic induction) can impact the synthesis of polyphenols in the complete part of jackfruit peel [40]. Other bioactive compounds in antioxidants can aid the chemical reactions in the human body and then help to offset the damages mediated by oxidation reactions. The pectin content in jackfruits is reported to contain 9 to 15% of dry weight and can make it a valuable source of polysaccharides [39,40]. The extraction of various compounds is also used in the textile, paper, and biofuel sectors, with rich sources of celluloses. This can provide alternative options to commercialized celluloses in the pharmaceutical industry [41,42].

*4.3. Waste towards Sustainable Energy and Biochemicals: The Attainment of Zero-Waste Technologies*

The harnessing of waste biomass for the derivation of high-value bioproducts presents a prospective avenue in the pursuit of sustainable waste governance for many fruits [43]. For example, diverse pathways for valorisation have been delineated for discarded orange peels, encompassing avenues such as biofuel production, biorefinery processes, pectin extraction methodologies, and the formulation of nutritive animal feed additives [44]. The residual matter arising from banana consumption emerges as a propitious and compatible

substrate, poised to fuel the advancement of environmentally benign bioenergy generation. This resource reservoir stands primed for exploitation, channelling its potential towards the establishment of regenerative processes that underpin a holistic circular economy paradigm [45]. The deployment of eco-friendly technologies warrants contemplation, mandating a systematic integration to enhance the inherent attributes of apple by-products, thus facilitating their valorisation into the foundational constituents of biochemicals and thereby leading to zero waste attainment [46]. The expansion of jackfruit plantation, owing to growing market demand coupled with increasing market size worldwide and the enormous waste generated by the jackfruit tree, nevertheless pose serious challenges for the effective disposal of waste and generation of sustainable energy and biochemicals following novel technological interventions.

### 4.3.1. Valorisation Techniques for Zero-Waste Generation

In current scenarios, a number of researchers are exploring the zero-waste generation concept and many strategies (i.e., various valorisation routes) are applied to achieve it. This was achieved using bread waste (BW) as the model development [42,47]. However, there are a number of challenges (i.e., technical processing steps) occurring for various kinds of food waste such as jackfruit waste also. Normally, any type of food waste hydrolysis, including jackfruit wastes, can be achieved via enzymatic hydrolysis with assisting physical and chemical agent processes. These approaches can generate monomeric compounds such as glucose/other sugars [47]. These sugars from jackfruit wastes can be utilized for the cultivation of suitable microbial systems with the capability of using carbon substrates and also the efficiency to metabolite into fuels sources such as ethanol/butanol/other bioproducts. One example is shown for the BW valorisation strategy with the use of a *Euglena gracilis* algae cultivation medium with a systematic evaluation [48].

This algal cultivation was found to be a future perspective and also economically viable for biodiesel sources production. In the context of valuable products other than biodiesel, other targeted compounds such as paramylon ($\beta$-1, 3-glucan) were synthesized from *E. gracilis* with a high productivity of 1.93 $gL^{-1}d^{-1}$, and this productivity was 24% higher than that of control strains. In this product synthesis, people applied the approach of zero-waste disposal and then bread waste residues (BWR) were hydrolysed using an enzymatic hydrolysis process, and it was later valorised into syngas [47,48]. This was offered by a greener pyrolysis process for BWR and carbon dioxide was used as a raw material for valuable products synthesis. $CO_2$-assisted valorisation is an attractive technique not only for efficient waste disposal, but in attaining climate neutral or zero carbon emission. In reports, several types of agricultural wastes (including jackfruit waste) have been found for utilization as potential raw materials for the generation of valuable products such as biofuels, biochar, or also biopesticides with briquettes or others [42,48]. Then, biochar can be mixed with soils and produce carbon-rich soils via contribution in carbon dioxide sequestration and soil fertility. Hence, jackfruit waste can be anaerobically utilized to produce biogas, briquettes, and biochar, which can improve crop production and serve as a carbon dioxide sink to mitigate the effects of climate change. The anaerobic digestion process for jackfruit waste has demonstrated economic viability for generating bioproducts [48,49].

The mechanistic functionality process for carbon dioxide ($CO_2$) mitigation is systematically described. In this process, $CO_2$ is reacted with volatile matter (VMs) that evolved from BWR, thereby helping to reduce the concentration of $CO_2$. It has been performed with proceeding to the oxidation of VMs. Further processes (such as consecutive gas-phase-reaction ~GPR) have been performed, exhibiting a critical role in enhanced CO formation [50]. The aim is to develop innovative delivery systems, such as nano-emulsions, for bioactive compounds derived from fruit and vegetable wastes, including jackfruit waste. Figure 3 illustrates complete jackfruit plants with their edible and non-edible parts that are later utilized as waste and then converted into value-added products with zero-waste generation.

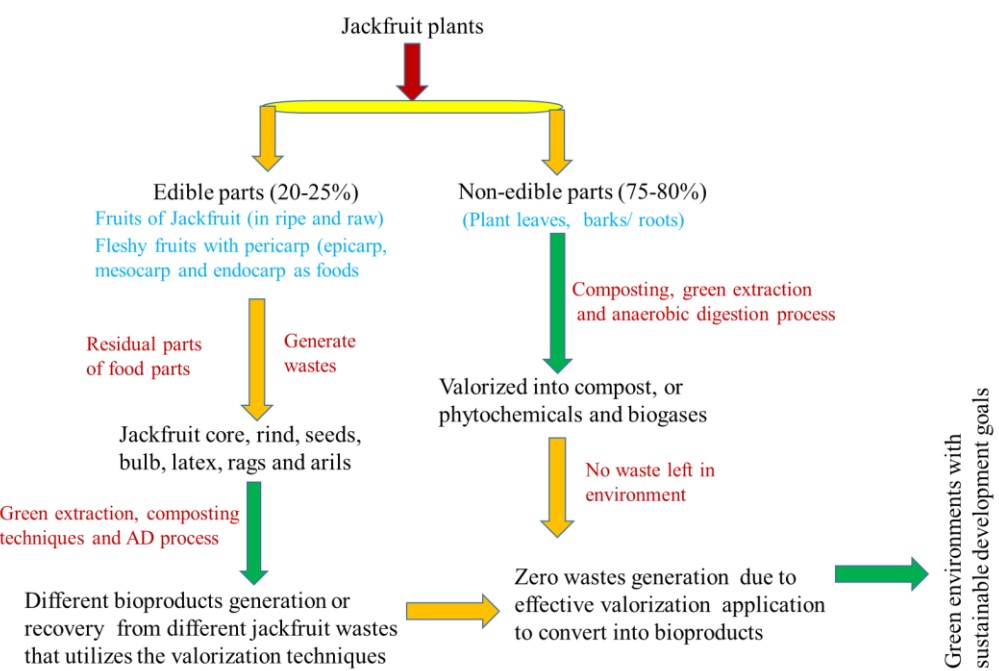

**Figure 3.** Schematic representation of jackfruit plant parts with its waste conversion into value-added products.

### 4.3.2. Advanced/Green Extraction Techniques for Bioactive Recovery

In valorisation processes, researchers have applied various types of novel extraction techniques such as enzyme-assisted, microwave-assisted, ultrasound-assisted, high-hydrostatic pressure-assisted, pulsed-electrical field, and super-critical fluid extraction methods to derive various bioactive compounds that are used in the food, pharmaceutical, cosmetic, and healthcare industries [40,42]. These non-conventional green extraction techniques have the potential for zero waste, thereby helping in sustainable energy conservation. Recent research has also shown that various biofuels can be produced through microbial fermentation. Some examples of biofuels (such as bioethanol up to 11–13%; biogas including methane via the AD process) with the help of *Saccharomyces cerevisiae* are produced through microbial fermentation [51,52]. Similarly, microbial fermentation has been effectively demonstrated to yield other valuable products such as bioactive compounds and other products [53]. Researchers have made sincere efforts towards developing novel and efficient techniques that can recover bioactive active compounds without any solvent contamination. Enzyme-assisted and pressurized liquid-assisted extraction have been explored for jackfruit wastes applicable to other wastes that can recover valuable products [54]. Microwave-assisted and ultrasound-assisted extraction techniques have been employed to derive pectin polysaccharides and antioxidant phenolics, respectively [55–57]. Pulsed electric field-assisted and supercritical fluid-assisted techniques have also been exploited for various natures of waste valorisation tasks that have recovered bioactive compounds at a high yield/concentration [48]. Among the various green modes of extraction techniques, their effectiveness can be varied with properties of the source matrix, its chemical structures, and also the process parameters/factors (solvent, pressure, time, or temperature [53,58].

Novel technologies can provide alternatives to conventional techniques for the extraction of bioactive waste from various food wastes, and these techniques are known to use water as a solvent rather than organic chemicals. These techniques have shown positive impacts on phytochemicals, scattering inside the cytoplasm [59]. Normally, the hydrogen or hydrophobic bonds present in the polysaccharide–lignin network/complex are found to be a big challenge due to exhibiting difficulty in the extraction of bioactive compounds [60]. Thus, these green techniques can be found to have a sustainable and eco-friendly nature

with the capability of achieving a higher yield of products compared to conventional ones. These techniques have received more attention in the last few years due to having more advantages [59,60]. Some techniques in the context of green/non-conventional mechanisms are discussed below.

In the enzyme-assisted extraction (EAE) process, the enzyme concentration, compositions, particle size, water-to-solid ratio, and also hydrolysis time play important roles and can influence the yield/concentration of bioactive compounds. Some compounds such as carotenoid extraction (from pumpkin waste) and anthocyanin from (*Crocus sativus*/grape fruit waste) have been achieved using the EAE process [61,62]. Some bioactive compounds such as phenolics (18–20 mgg$^{-1}$) have been extracted from grape mare seed wastes with the help of pectinase enzyme activity [60]. Other efforts have been made towards antioxidant phenol extraction from apple pomace by using the commercial enzyme Pectinex Ò$^®$, and then phenolics recovery (87%) from grape residues using Celluclast Ò$^®$. The EAE-based technique is very effective in enhancing the recovery of enzymes such as pectinase, cellulases, and pectinases from jackfruit waste [63,64].

The second extraction method is known as ultrasound-assisted extraction (UAE), and it has a number of uses and benefits, including a larger yield, desired quality, and a straightforward procedure with minimal impact on the environment. The UAE approach uses the ideal frequency range (20–2000 kHz) and is well renowned for being both straightforward and inexpensive. It can be effective at two different aspects, such as diffusion over the cell wall and washing the contents following cell disintegration [65,66]. Due to the wave creation of matrix expansion and compression, the UAE operating mechanism causes/generates a cavitation phenomenon. The desired compounds are then extracted by causing the cell membrane to become permeable [66]. Researchers have investigated the many processes of the UAE process, including the acceleration of mass transfer, the disintegration of the particles, and the improvement in solvent accessibility. For this technique, samples of liquid–liquid or liquid–solid processing are frequently employed. Pressure, temperature, frequency, and sonication duration are other parameters that might affect the UAE process [65–67]. Recent research on UAE has revealed the widespread use of this procedure and showed its effects on yield and compound characteristics. Some excellent instances of UAE frequency have been documented for energy at or above 20 kHz, which changes the physical–chemical characteristics of phytochemicals by causing the production of free radicals [67]. Figure 4 establishes the sophisticated processing methodologies employed for the extraction of bioactive compounds.

Some efforts towards tannin extraction from Avaram shell have been reported with the application of the UAE technique and it uses 100 W of power. The UAE-based process has shown its impact through the improved yield (160%) of tannins at 100 W. This improvement in the yield of tannin has been explored and was found due to the improved mass transfer of cell components and a way of leaching of tannin via this power [68]. Other bioactive compounds, such as a good yield (caffeic acid ~64.3 μg/g, ferulic acid ~1513 μg/g, and p-coumaric acid ~140 μg/g) of phenolic acids via the UAE technique, have been reported with a better improvement compared to conventional techniques (maceration extraction) for the same bioactive compounds from the same wastes [69]. Temperature and prolonged time impact on UAE are found in the form of a low yield of phenolic compounds from citrus peels. Some comparative studies have been performed on the maceration and UAE technique performances in terms of consumed/required time period for the extraction of bioactive compounds. A reduced time period (1 h) was found for the UAE technique compared to maceration-assisted extraction (72 h) for phenolic compounds from *Punica granatum* fruits. The extraction of polysaccharides using the UAE technique resulted in a good yield, proving it to be an efficient technique [67–69].

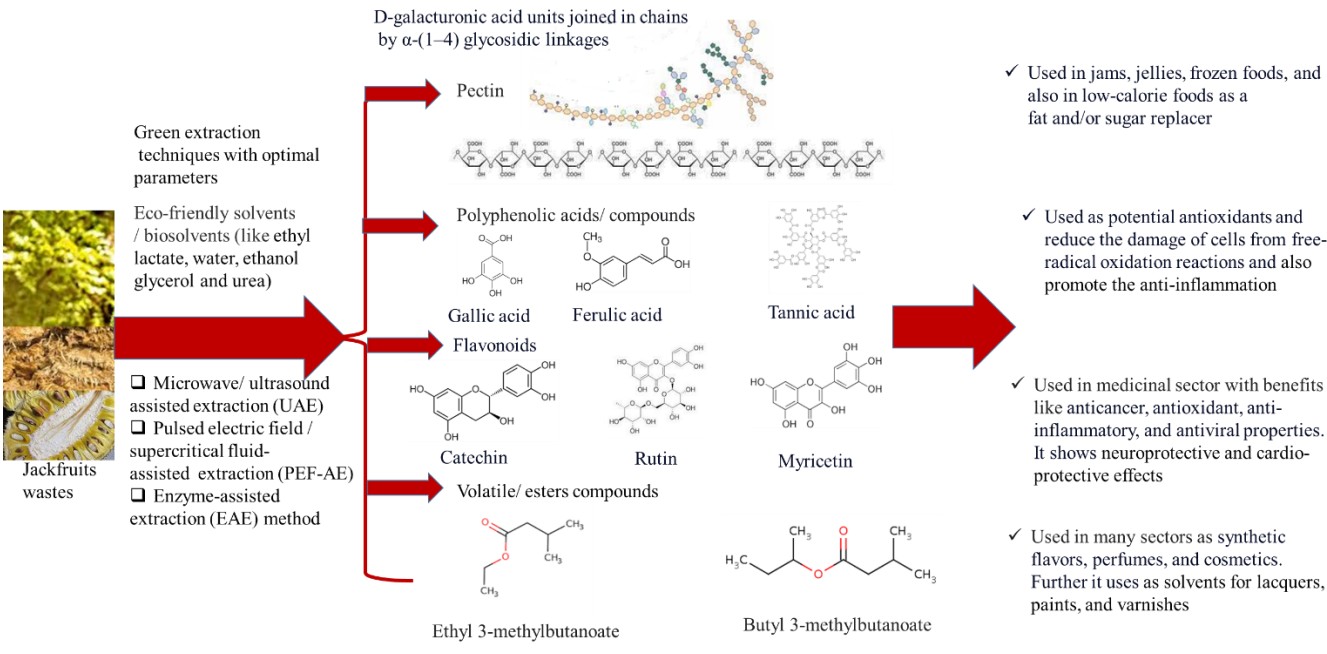

**Figure 4.** Green extraction for better yield of bioactive compounds from jackfruit waste.

Yet another green extraction technique is known as pulse electric field-assisted extraction (PEF-AE), a non-thermal process that allows a direct current to produce, which helps produce a better yield of bioactive compounds. In this technique, the application of a high voltage current/pulse is passed through the materials that are kept/placed between two electrodes for a short time period (in the range of microsecond to millisecond periods). During the passage of the electric current through the suspension of cells, this can influence the cell structures to be destroyed and then molecules can separate with respect to the applied charge [70]. This technique can function in a batch and also in continuous modes. For this technique's performance, various factors such as field strength, energy, pulse number, temperature, and material properties can affect/influence the yield of the extraction and can work based on designing a process for a better performance. The application of PEF-assisted extraction can be performed with phenolic compounds and also the anthocyanins extraction process from different wastes [71,72]. PEF-assisted extraction with maceration on grape skin can be applied for the stability of the compound during vinification and can reduce the time of extraction. Some studies have been conducted on untreated control samples along with PEF-treated samples with the maceration extraction process/technique [70,71]. This has shown improved colour and also anthocyanin content/yields with enhanced polyphenol contents in wastes such as jackfruit [72]. Further impacts of PEF treatment can be found in the wine-making process, with a reduced maceration time and also an improved wine quality [73–76]. The cytomembrane in the plant tissue's cell walls can influence the movement of intracellular material between cells. This approach for extracting bioactive substances is intriguing because it may trigger the cytomembrane in the tissue to disintegrate, which changes its permeable characteristics, as well as increasing mass transfer across the cells, leading to higher yields [74,75].

Advanced/green extraction techniques utilize electromagnetic radiation and this is transferred in the form of waves in a frequency range (300 MHz–300 GHz) with the common use of a frequency of 2450 MHz. This can be found to be equivalent to 600 to 700 W of energy and this energy can be absorbed during the passage of the microwave through a suitable medium. This medium converts it into thermal energy by facilitating the processing [76,77]. Some bioactive compounds such as flavonoids from *Terminalia bellerica* plant have been reported by using this technique and microwaves in the MAE

technique can result in a maximum yield (83%), with higher yields while compared to those of conventional techniques (flavonoid yield ~64%). The MAE technique uses microwave energy for heating the solvents with samples and is also influenced by operating conditions such as temperature and feed ratio on flavonoid yield. The same extraction was applied for hesperidin compound extraction with a better yield (48%) from the skin of *Citrus unshiu* fruits [78,79]. During this technique, the extraction temperature showed a high impact on the bioactive compound yield, and at 140 °C, it showed a decreased hesperidin quantity/content due to the interference of the other solubilized substance. This compound influence has been found in the inhibition of hesperidin crystal [77,79]. Other conditions impact waste matter maturity level in the case of peels, and matured peels have yielded fewer hesperidin contents compared to immature peels (more than three times). Some impacts of power in the MAE technique have been reported and this was found for phenolic compound extraction from chokeberries. A better yield of phenolic compounds (420.1 equivalent mg gallic acid/100 g chokeberries) was reported at 300 W for 5 min [80,81]. The extraction of silibinin from *Silybum marianum* waste with the help of the MAE technique is discussed with a better yield (97.3%), and this was higher than that of conventional approaches. A similar study was performed on this technique's efficiency and it was reported for phenolic compounds from apple pomace waste [82]. The compound extraction yield from this technique is influenced by several factors/parameters such as solubility, dielectric constant, dissipation factor (d), and solvent nature. The higher recovery of flavonoids using MAE (up to 74%) is reported and it is better than that of the traditional recovery/extraction process (up to 70.5%), with proof of an efficient process [83,84].

During the supercritical fluid extraction process (SC-FE), the desired compound extraction is carried out by using the solvent above the critical point (CP), and this CP can be found as a specific temperature (Tc) or pressure (Pc) point. However, above the CP, gas and liquid cannot exist as separate phases [85]. At the CP, fluids/solvents can exhibit the liquid (in terms of density) and salvation power/gas (viscosity, diffusion, and surface tension). These properties can facilitate a higher yield of bioactive compounds within a short time. In the case of supercritical fluid extraction (SFE), there is a need for a good mobile tank (consisting of a $CO_2$ pump), solvent vessel, oven, controller, and also a trapping vessel [85,86]. Now, most bioactive extraction is achieved through the application of green technologies compared to conventional methods, and in this context, supercritical $CO_2$ extraction is discussed with a better yield of naringin from citrus paradise. This approach uses ethanol as a modifier (14% by wt.) at the same process conditions such as a 58.6 °C temperature and 9.5 MPa pressure [87]. This technique is applied for phenolic compound extraction from rice wine lees wastes, with the uses of Soxhlet extraction (SE) and SFE. It was compared with the yield of bioactive compounds with a reduced time period extraction (1 h compared to 6 h in the case of the traditional extraction technique), with less ethanol needed and a better yield of phenols (43%) [88]. In the process of SFE, carbon dioxide is a common solvent used in food sector tasks and it is safe with easily attainable critical conditions (at 30.9 and 73.8 bar) for food processing. Some major limitations such as a low polarity are disclosed, but they can be improved by using polar solvents such as methanol, ethanol, dichloromethane, and acetone [87,88]. These can work as modifiers with the capability of improving their solvating power and also enhancing their extraction efficiency with minimum/no interaction between analytes and matrices. Some parameters such as a low diffusibility of the solvent into the matrix, an extended extraction time, high-pressure requirements, and expensive infrastructures can also be found as some challenges to this technique [86,87]. Further, the consistency and reproducibility during the continuous process can be found as some more limitations of this extraction technique and these can prevent the scalability of this technique [87,89].

### 4.3.3. Microbial Fermentation for Jackfruit Waste Conversion

For the better utilization of jackfruit waste, it needs several types of effective pretreatment processes, and physical, chemical, and biological processes can be applied to disrupt

its complex organic matter. These strategies help in the valorisation of jackfruit waste into value-added products, including fuels (bioethanol or biogas) and other bioproducts (bioplastic, feeds, or functional food additives) [90]. From jackfruit waste, nowadays, bioenergy production and promotion have also been carried out by several research groups and jackfruit waste can be utilized as a renewable resource. It is also an eco-friendly and cost-effective process for generating alternative fuel options to fossil fuel [91]. In recent years, the efficient bioconversion of jackfruit waste into several types of fuel sources has been performed with the help of the microbial fermentation process by using different microbes such as bacteria, yeast, and fungi [84,85]. Yeast (such as *Saccharomyces cerevisiae* for ethanol), bacteria (such as *Methanosarcina barkeri* and *Methanococcus maripaludis* for methane), and fungi (such as *Rhizopus oryzae* MNT 006, *Aspergillus oryzae* MNT for ethanol) have been reported for biofuel production [24,49,92]. These efforts can help with the reduction in environmental pollution and also help with bioremediation processes (i.e., several toxic dye removals) in water sources/aquatic environments that are contaminated with dye colour. Some review papers have addressed colour dye removal from water bodies with the help of jackfruit waste. This effort can solve several serious ecological problems in jackfruit-producing nations [91,93].

Several reports have claimed a waste generation quantity of jackfruit in the range of 5 to 7 kg wt. per fruit, and its potential for conversion into a wide range of bio-products such as biofuels, animal feeds, or bioactive components for use in the bakery and packaging material industries. In the process of waste (including jackfruit waste) material hydrolysis, physical, chemical, and biological pre-treatments processes are applied. These processes help the conversion of waste into simple sugars in final production synthesis under the effort of valorisation [94]. The utilization of jackfruit waste can result in several fuel product syntheses/productions. These can be achieved by the application of pre-treatment and extraction steps via the valorisation of waste biomasses in an effective and successful manner. In recent years, a number of valorisation technologies have been developed for jackfruit waste hydrolysis and, later, these have been used in the conversion of desired products via the utilization of several bioprocesses/green extraction steps. Thus, they help in promoting sustainable product utilization in the biorefinery and also the bioeconomy [13]. Normally, jackfruit is reported to contain 70–80% non-edible parts and out of this quantity, 60% of jackfruit wastes are the outer rind, perianth, and central core parts. A number of analyses for the biochemical composition of jackfruit waste have been performed, with the utilization of these wastes for the recovery of health benefit products [13,94]. The peels of jackfruit waste are good sources of protein, cellulose, and pectin and then seed waste is a good source of carbohydrate (76%), protein (18%), and lipid content (2%). In the context of bioenergy production from jackfruit waste utilization, several types of pre-treatment are applied as crucial steps with a conversion capacity from complex forms of organic matter into simpler ones. The pre-treatment step in the complex organic matter conversion process helps in the enzymatic reactions of hydrolysis during saccharification processes, and this step can ensure the simple sugars' form for fermentation with the help of different microbial agents [95]. A good example of jackfruit waste conversion is found in the ethanol production/extraction process. This process utilizes low-pressure and also high-intensity ultrasound processes that affect the compositions and functionality of the isolated proteins from jackfruit seeds [13,90]. Jackfruit waste has undergone several processing tasks with a variety of physical methods to develop its valuable products, and some of these are irradiation, microwave processing, super-critical fluid, and high-pressure processing extraction, as common and advanced processing methods [95,96].

In an effort toward the hydrolysis of jackfruit wastes, several approaches/methods of chemical treatment have been applied in acidic or alkaline solutions (low to high concentration) at temperatures of 130 °C and 210 °C via mixing the waste matter. During the chemical-agent-assisted pre-treatment process, waste matter can be converted within a few minutes to a few hours to obtain fermentable sugars depending on the pre-treatment conditions [97,98]. Many researchers have applied these processes for methane production,

efficient energy generation, and also environmental benefits using jackfruit peels. The effect of methane generation via the pre-treatment of jackfruit waste with 5% alkaline hydrogen peroxide (AHP) has shown an enhanced methane yield, resulting in a higher biodegradability (up to 70%) when compared to untreated waste matter [99]. Some studies too have reported the potential of the annual energy production from jackfruits, treated with a 5% AHP solution. Further, alkaline extraction techniques have been applied for the isolation of starch from jackfruit seed waste, and this technique was found to extract 18% starch from seeds [98,99].

Conventional chemical and physical pre-treatment processes require the costly investment of reagents, machinery, and energy before applying biological pre-treatments for cellulosic and lignocellulosic matters in enzymatic saccharification processes [17,100]. During the biological treatment process, many living agents such as fungi/bacteria are used. They utilize less energy and are thus more eco-friendly processes. Many microbial agents are known to possess cellulolytic and hemicellulolytic activities that can be utilized for jackfruit waste utilization [2]. Among several microbial systems, *Saccharomyces cerevisiae* is applied for ethanol production from jackfruit waste—ethanol contains 35% oxygen content and can be utilized for the burning process in the production of a lesser quantity of nitrogen and also particulate matters in the gasoline combustion process [2,99]. Figure 5 discusses the pretreatment approaches for hydrolysis and also bioenergy development.

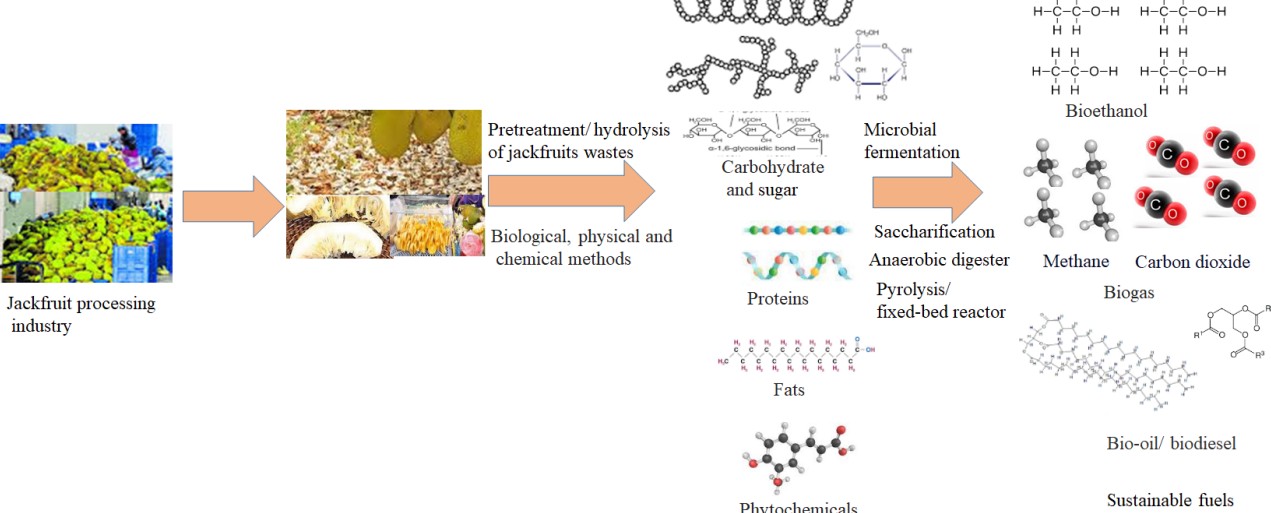

**Figure 5.** Jackfruit waste utilized for bioenergy task.

*4.4. Value-Added Products*

Nevertheless, huge quantities of various types of fruit and vegetable waste are generated due to different the processing activities in the food sector every year. These form excellent sources of different natural, valuable components such as biofuels and polyphenols and are utilized for different needs of human life [101]. Green extraction technology, often known as the non-traditional approach, has been a significant subject of research in recent years [102]. These green technologies have replaced previous traditional methods due to their high yield, shortened process time, products of superior quality, and low waste creation [98,100]. During the pre-treatment process, a disruption of the hydrogen bond is also found and can occur easily after the addition of a chemical solvent, and then it is heated at a high temperature. This can cause the dipole rotation of molecules and the migration of ions. These changes in matter can occur with the diffusion of the solvent and can lead to the dissolution of the components [103]. Another mechanism during the pretreatment process is reported as an evaporation of the moisture within cells, which can induce high pressure on the cell wall and then change the physical properties of the materials [104]. In turn, this leads to a modification in the porosities of biomaterials, resulting into an increased

penetration of solvents with an improved yield of biomaterials [105,106]. Various types of bioproducts from jackfruit wastes are discussed in the below section.

4.4.1. Bioethanol

In the context of value-added product recovery from jackfruit waste, bioethanol production from enzymatic hydrolysis and the microbial fermentation of cellulosic biomass, including jackfruit wastes, has been discussed as a cost-effective and eco-friendly technique/approach [107]. In this fuel synthesis process, jackfruit wastes such as peel material are utilized as a potential feedstock and this waste matter contains high quantities of carbohydrate contents. A number of studies have been conducted for saccharifying jackfruit rind matter via the application of recombinant enzyme endoglucanase from *Bacillus subtilis* strain MUS1 microbes at a temperature of 50 °C and pH of 5.0 with 15 mg/mL of substrate ratio/quantity [108]. This recombinant enzyme from *Bacillus subtilis* can help in the generation of more quantities of sugars during the saccharification process of jackfruit waste [51,107]. From different types of waste, bioethanol is produced as an alternative energy source. These wastes are good sources of natural products such as carbohydrates and these are utilized as a potential feedstock for ethanol production with the hydrolysis and fermentation process. In this context, jackfruit seed is the best feedstock with rich sources of carbohydrates [108,109]. A number of research studies have determined the effect of pH on carbohydrate hydrolysis that can be utilized for bioethanol production. For the hydrolysis of the carbohydrates from jackfruit seed, effective pre-treatments with a pH-dependent process have been performed via the application of a separate fermentation hydrolysis (SHF) process [109]. This process used a sulphuric acid solution as a hydrolysing agent. Later, the fermentation process with the help of the *S. cerevisiae* strain was used in a fermenter vessel at different pH values (such as 2, 3, and 5) over a 70 h period. From this experiment's results, it was claimed that an optimum glucose content (75%) and pH (3.0) supported the high concentration of bioethanol (58%) in the fermentation broth [108–110]. From this research work, the fermentation stage's role has been discussed and it was found that a high concentration of glucose can push a high concentration of bioethanol with a linear relationship [110]. This published paper talked about the glucose concentration and high ethanol concentration from jackfruit seed waste, which have shown a high potential for feedstock for bioethanol biosynthesis at a cheap price [108,110].

Bioethanol production from fermenting raw matter such as jackfruit waste is reported with suitable microbial systems. During the fermentation process, the ethanol production of jackfruit waste is reported with the application of the *S. cerevisiae* strain. This yeast strain has shown a high capability for ethanol production, which is due to its natural adaptation properties and also a highly tolerant sugar/ethanol/chemical inhibitor [111]. Ethanol can be synthesized by petroleum products or the fermentation process. In the biological process, cellulose or hemicelluloses from jackfruit waste are hydrolysed and then utilized for the ethanol production process. Most lignocellulosic biomass, including jackfruit waste, is a rich source of carbohydrates, and thus it needs an effective pre-treatment process for fermentative sugars that can be used in ethanol production. Later, product separation and purification processes are also needed to obtain pure ethanol [108,111]. For this bioethanol production, jackfruit peels were selected as potential substrates, and then an enzyme/microbial system (*S. cerevisiae*) was used for obtaining the fermentative sugars and ethanol. Thus, ethanol now can be present in alcoholic drinks, with more uses of this yeast in the bakery industry and fermented food and alcoholic drink preparation/production [111,112]. Some efforts have been made towards jackfruit straw waste and it has been processed with fermentation as a starter substrate. From this process, bioethanol was separated via a distillation process. The starter mass and fermentation time were checked to achieve the maximum ethanol yield with the utilization of yeast such as *S. cerevisiae* and urea as N-nutrients [112].

An optimum yeast mass (40 g) of *S. cerevisiae* with a distillate volume of 13.6 mL has been found after 96 h of fermentation time, with a distillate/ethanol yield of 15.2 mL.

However, under the optimal conditions for fermentation, the volume of the bioethanol distillate was found to be 30 mL with a distillation temperature of 70–80 °C. From this approach to ethanol production, its refractive index (1.354), density (0.367 g/mL), and boiling points (71–72 °C) were reported [113]. In another report, bioethanol production was discussed from the utilization of Sri Lankan rotten fruits (without skin), including jackfruit waste. This ethanol was produced in a batch process with an optimization of the fermentation process parameters [112,113]. In the optimization of the fermentation process, some optimization techniques such as the Genetic Algorithm (GA), Response Surface Methodology (RSM), and also Particle Swarm Optimization (PSO) were discussed. During the bioethanol production, different overripe fruits were taken and these fruits were jackfruit, papaya, and banana, with the use of two microbes under three fermentation conditions. During these experiments, maximum ethanol yields were reported with the RSM (13.4 vol. %), GA (13.4 vol. %), and PSO (13. 36 vol. %) with the use of banana variety fruit fermentation and a *Pseudomonas mendocina* microbial strain (ratio~ 1:1), pH (5.1) and temperature (35 °C) [114,115].

4.4.2. Biogas

Biogas generation from different types of jackfruit waste is a good effort for sustainable fuel sources. In this context, the potential of biogas production is found from different fruit wastes such as banana peels, jackfruit waste, and pineapple waste, and these have been used for co-digestion processes with cow dung to provide alternative energy sources [116]. During these experiments, substrates from each fruit waste were implemented into the co-digestion process with varying ratios (0%, 25%, and 50%) of cow dung. This was performed in laboratory-scale anaerobic digesters (up to a capacity of 250 mL) and was then run for 30 days to generate or produce the biogas from different fruit wastes (jackfruit— 82.3 mL; banana fruit—189 mL; and pineapple fruit waste/peel—262 mL) [117]. In another experiment in this study, jackfruit waste, pineapple waste, and banana peels were co-digested with 25% cow dung, and the biogas production from these fruits increased by two to three folds [117,118]. Cow dung is a very effective medium for biogas generation. Studies have shown that a 50% cow dung ratio with these fruit wastes improves the biogas yield by two folds. From these reported results, a mixture of jackfruit, banana peel, and pineapple peel can be found in a much better biogas production yield and can help in the energy supply chain process for our daily needs [118]. During the biogas generation experiment, some efforts were made towards experimental design, digester set-up, and the volume and biogas composition determination that were produced from the jackfruit waste, banana, and pineapple peels with the cow dung. The biochemical methane potential (BMP) assay protocol was applied for the anaerobic digestion process [119]. An evaluation process for the biogas quality attributes was performed for a process that used jackfruit waste, banana, and pineapple peels with cow dung in a batch digestion process. In this experiment, an anaerobic system was found to have a 500 mL capacity and was submerged in a 20 L temperature-regulating water bath and 250 mL measuring cylinder for the generated biogas measurement task via the displacement method. The temperature of the water bath was maintained at 36.5 °C [116,119].

In the context of value-added product recovery, the utilization of jackfruits has been reported and discussed for the production of biogas, biochar, and briquettes from jackfruit waste. In many developing countries, huge potential is found due to the greater quantity of organic waste accumulation that can then be utilized for conversion into many types of fuel sources such as biogas generation [49,51]. Waste organic matter in rural areas due to small holder farmers can be generated and then these can be utilized for different natures of biofuel sources. In this context, biomass wastes such as jackfruit waste can be managed to produce bioenergy with a mitigation of the GHG (greenhouse gases) emission quantity via a well-managed way [120]. Now, the decomposition of organic matter can be minimized via the generation of biogas, due to greater quantities of agricultural waste such as jackfruit waste being utilized as cheap raw materials for the production of bioproducts such as

biofuels, biochar, and biopesticides with briquettes and others [121]. Biochar production from waste matter is a good effort, and then biochar can be mixed with soils and help to produce soils rich in carbon with a contribution to carbon dioxide sequestration and soil fertility [122]. Some papers have focused on jackfruit waste utilization for biogas production from an anaerobic digestion process with biochar and briquette production. From the anaerobic process for jackfruit waste utilization, biogas can be produced [117,119]. It needs a high temperature for jackfruit waste decomposition tasks to help in biochar production. This research effort can help to produce various product synthesis information, with help in the mitigation of climatic changes and also carbon dioxide sinks in soil [49,122]. Figure 6 discusses the jackfruit wastes with different microbial/chemical transformation approaches for many products syntheses to support zero-waste generation.

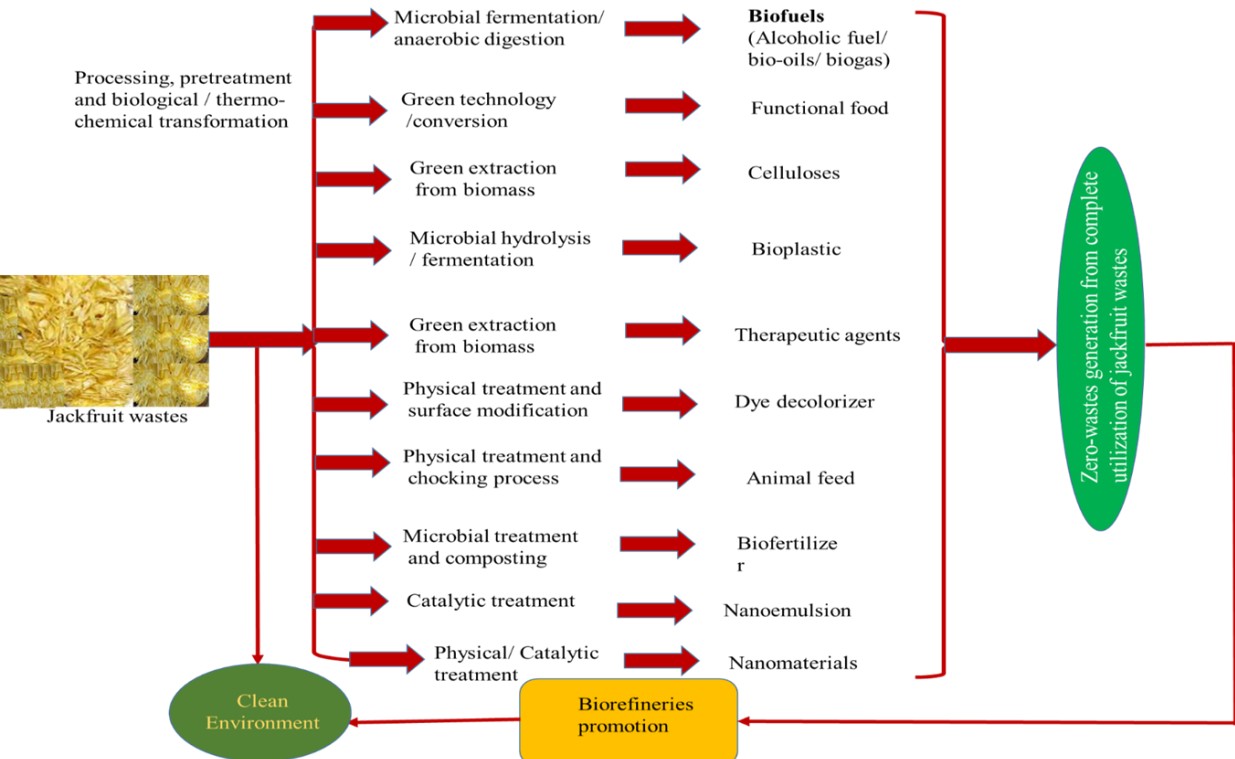

**Figure 6.** Jackfruit wastes promote zero-waste generation and biorefinery promotion.

### 4.4.3. Bioplastic

Bioplastic production from the *Bacillus megaterium* strain JHA has been reported, with a high capacity of this microbial stain. This microbe was isolated from oil-contaminated soils and then tested in the laboratory on glucose substrate consumption, resulting in a high quantity of polyhydroxyalkanoates (PHA) accumulation in this microbe [123]. This plastic was biosynthesized by *B. megaterium* with the utilization of jackfruit-waste-like seeds, with some inorganic or organic matter/compounds. This waste matter was found to be suitable for a high accumulation of PHAs inside microbial cells. Some additional efforts were made toward a characterization of the bioplastic layer that formed inside the cell due to the PHAs [124]. Several viable applications of PHAs have been found in medical science and cosmetic and pharmaceutical product syntheses. For the best synthesis of PHAs, advanced research is needed with an optimization of the process parameters, and this could promote huge quantities of PHAs from cheaper feedstock such as jackfruit wastes. This waste matter is a good source of carbon and nitrogen for finding an economically viable process for commercial scale [123,124]. A further PHA product is biodegradable plastic and it has shown an effective and durable process. In an effort toward jackfruit waste hydrolysis, biological pretreatment is a promising and eco-friendly process for complete conversion

into fermentable sugars [125]. Biological pre-treatment needs optimal process parameters to achieve the complete hydrolysis of waste biomass. The effect of pre-treatments such as physical and chemical processes can enhance the impact of the enzymatic conversion of waste biomasses into simple ones, achieving bioproduct synthesis [124,125]. Table 1 discusses more examples of bioenergy/other value-added products.

**Table 1.** Different value-added products (i.e., biofuel, bioplastic, enzymes, pectin, and nano-emulsion) synthesis from jackfruit waste conversion via valorisation techniques.

| Jackfruit Wastes | Value-Added Products | Application | References |
|---|---|---|---|
| Jackfruit wastes with its different parts such as peel/skins | Bioethanol, biogas, bioplastics, and so on generated. | Biofuel is greener than fossil fuels. Processed peel cleans dye-contaminated aquatic environments. | [8] |
| Durian shell and jackfruit peel | An increase of 103.8% in methane output and 69.8% in biodegradability. | Produces high levels of sustainable energy and fuel ($2.0 \times 10^9$ MJ/year), while simultaneously decreasing coal usage ($6.8 \times 10^4$ tons/year) and cutting emissions by $2.2 \times 10^{10}$ particulate/year. | [10] |
| Jackfruit outer rind | Used as substrate for producing recombinant endoglucanase from *Bacillus subtilis* MU S1. | This enzyme helps in the highest saccharification process (33.4%) from jackfruit outer rind at 96 h of incubation. | [12] |
| Jackfruit peel waste | Bio-oil upgrading is performed by sub/super-critical fluids, solvent addition, and steam reforming. | The application of bio-oil can substitute fuel for a commercial or industrial burner. | [13] |
| Jackfruit seed starch was plasticized | Used to produce starch-based bioplastics. | Four distinct kinds of bioplastics were manufactured in order to investigate the influence of the plasticizers and characterise the features of the bioplastics that corresponded to those plasticizers. | [26] |
| Utilization of jackfruit peel | There have been reports of the use of an adsorbent made from jackfruit peel. | Jackfruit peel adsorbent can biosorb maximally 232.55 mg/g of methylene blue (MB). | [27] |
| *Artocarpus heterophyllus* Lam. seed powder extract (ASPE) | Green production of silver nanoparticles from silver nitrate in water. | ASPE may be utilized to synthesize AgNPs for nanomedicine in the future. It is eco-friendly and harmless. | [33] |
| *Artocarpus heterophyllus* peels | Green synthesis of iron nanoparticles is reported. | Iron nanoparticles were highly catalytic, removing 87.5% in 20 min at 318 K. | [36] |
| *Artocarpus heterophyllus* (jackfruit) peel | Bio-based coagulating agent. | The extract from the peel has the potential to serve as a bio-based coagulating agent alternative that is useful in the pre-treatment of wastewater. | [40] |
| Jackfruit seed powder (JSP) | The surface of JSP is used. | Novacron blue textile dye can be decolorized using this substance. After a contact time of 60 min, the surface of JSP has been observed to adsorb 73% of Novacron blue. | [93] |
| Jackfruit waste feedstock | Biogas production was improved by chemical catalysts with maintaining the pH and C/N ratio. | Biogas is produced via decreasing the digestion time with improving the efficiency of digester unit by using jackfruit waste as raw material. This raw material contains significant quantity of fibre with small proportions of glucose. | [94] |

**Table 1.** *Cont.*

| Jackfruit Wastes | Value-Added Products | Application | References |
|---|---|---|---|
| Jackfruit waste together with peels (JP) as well as seeds (JS) | Bioenergy has optimal physicochemical, bioenergy indicators, combustion, and emission properties. | The bioenergy yields for JP and JS were 2.5 and 0.9 ha$^{-1}$ year$^{-1}$ (dry basis), correspondingly. Low concentrations of CO, $CO_2$, and $SO_2$ may be released. | [17] |
| Jackfruit straw waste | Used as raw material for making bioethanol. | The amount of bioethanol distillate that could be produced under ideal circumstances was 30 mL. This research employed a distillation temperature range of 70 to 78 °C. | [104] |
| Jackfruit (*Artocarpus heterophyllus*) stone waste | Ethanol from these waste is found and it is used as renewable fuels. | A quantity of jackfruit flour was subjected to hydrolysis by the addition of 0.3 to 0.7 mL of alpha-amylase and 0.2 to 0.6 mL of glucoamylase. The resulting mixture was then subjected to a fermentation process lasting between 3 and 6 days. The yield of ethanol obtained from this process was found to be between 11 and 13%. | [51] |
| Renewable energy from jackfruit's seeds | Bioethanol (57.94%) is reported. | The fermentation process for ethanol production is used with *S. cereviceae* with a variation in pH values for 70 h. | [108] |

### 4.4.4. Bioactive Compounds

Pectin used in the food sector and pharmaceutical/cosmetics sectors is reported to serve as various agents such as emulsifiers, binders, and stabilizers, performing various functions [41]. Polysaccharide sources in jackfruit peels can help in the human diet and maintain good immunity activity with some protective functions such as against cancer, blood sugar, ulcers, and bad cholesterol [42]. In the pharmaceutical industries, pectin is utilized as a bonding agent for various formulations of pills and also multi-purpose delivery. In the food sectors, the use of jackfruit peel is employed for papers, paints, optics, and also environmental remediation with biofuel sectors. In the case of the phenolic compound yield from orange peel, an improvement in this yield (15%) was achieved with PEF at power of 7 KV/cm [72].

Reports on greater yields of phenolic compounds (102.9 mg GAE/100 g food waste) and flavonoid compounds (37.6% QE/100 g FW) from various wastes, such as onion waste, are described in this approach, with a superior yield improvement (2.2 and 2.7 times, respectively) compared to control samples. The effects of electric field intensity and extraction duration on phenolic and flavonoid compound yields have been demonstrated [73]. Nowadays, efforts are being made in the food processing sector toward producing nanoemulsions, which can be useful in the delivery systems for various bioactive substances capable of performing various functions. These actions boost bioavailability, regulate ingredient discharge, and alter product texture, as well as preserve the substance from degradation [105,106]. In this technique, intensive research is going on to make it more effective in terms of a broad range of bioactive extractions from any type of wastes, including jackfruit waste. For the nanoemulsion process, the delivery system can proceed for bioactive compounds with an understanding of their specific functions, especially in terms of food-based delivery systems and bioavailability in the human body [104,106]. Some recent studies have been performed to understand the mechanisms for achieving a desirable bio-accessibility, metabolism, and absorption of the encapsulated compounds, and these can help in altering in their properties in the gastro-intestinal tract [106,107].

Further studies have been conducted on the loading capacity of bioactive compounds that are in encapsulated form, such as nanoemulsions, exhibiting the better release properties of bioactive compounds [105,106]. The cultivation of jackfruit plants in the world is

good and the potential sources of valuable biomaterials and wastes from jackfruit plants are a good source of carbohydrates, fats, proteins, and also phytochemicals [104–106]. In the case of bioactive compounds from the wastes of fruits/vegetable sources, these have shown a positive impact on human health via contributing to the modulation of metabolic processes and also cellular activities [101]. Some bioactive compounds have shown properties such as antioxidant, anti-cancer, anti-inflammation, and anti-allergenic. Some compounds contribute to anti-atherogenic activity and these properties of bioactive compounds can depend on their pathways and also their bioavailability in the human body. In terms of the categories of bioactive compounds, some are hydrophobic in nature and they have shown less bioavailability in the human body [102]. In this context, some efforts have been made toward technological advancements such as nano-emulsion applications. This effort helped in enhancing their stability and functional properties. Bioactive substances can be obtained via traditional and non-traditional methods, each with their own set of benefits and drawbacks [101,102]. Table 2 shows the bioactive compounds with health benefits from jackfruit waste.

**Table 2.** Health benefits from bioactive compounds that are recovered from jackfruit waste matters.

| Jackfruit Wastes | Bioactive Compounds | Health Benefits | References |
|---|---|---|---|
| Jackfruit skins, leaves, and barks | Vitamins, minerals, and phytochemicals | The substance under consideration exhibits various properties such as anticarcinogenic, antimicrobial, antifungal, anti-inflammatory, wound healing, and hypoglycemic effects. | [2] |
| Agro-residues from jackfruit plants | Nanocapsule use increases their bioactive efficacy | Bioactive in micro- and nanoencapsulation forms and enhances target site delivery in human body with more benefits. | [4] |
| Jackfruits are known for their prickly outer bark and axis | Flavonoids, stillbenoids, morin, artocarpin, dihydromorinm, and cynomacurin | Because of their bioactivity, these chemicals have the potential to be developed into nutraceuticals with antioxidant characteristics. | [6] |
| Leave and stem bark extract of *Artocarpus heterophyllus* | Extract is dominated with tannin and saponin | The methanol extract of *Artocarpus* stem and leaf bark has antibacterial and antioxidant properties, and the bark may be used topically as a peel-off mask. | [7] |
| Jackfruit peel | Functional food additives and pectin materials | These compounds are manipulated for food ingredient applications, providing health benefits for gastro-intestinal tract. | [9] |
| Spine, skin, and rind of jack fruit | Polyphenols as well as flavonoids | Crude ethanolic extracts undergo evaluation for their anti-inflammatory potential. | [22] |
| Fruit peel of jackfruit | Flavonoids, and also presents some phenolic compounds | Various phyto-constituents can be used in different additives for human use with more health benefits. | [29] |
| Jackfruit seed extracts in three different solvents: methanol, hydroalcoholic, and aqueous | Alkaloids, flavonoids, terpenoids, and so on | These extracts have shown antioxidant, anti-inflammatory, and antibacterial activity. | [31] |
| *Artocarpus heterophyllus* J33 rind parts | Proto-catechuic acid (PCA) antioxidant activity depends on its temperature: 25 °C, 4 °C, and −18 °C | The extract's antioxidant activity was maintained because PCA is so stable. It has antioxidant properties and might be used in food and dietary supplements. | [51] |

**Table 2.** *Cont.*

| Jackfruit Wastes | Bioactive Compounds | Health Benefits | References |
|---|---|---|---|
| Seeds of jack fruit | The different solvent extracts showed the presence of fats, phenols, and flavonoids | Acetone extract demonstrated the greatest antibacterial activity towards *Staphylococcus aureus* and the greatest antifungal activity against *Aspergillus flavus*. | [101] |
| Jack fruit seeds | Total phenolic compounds in this seed extract confirmed by different screening tests | Total phenolic substances are identified as antioxidants with radical scavenging action. | [102] |

## 5. Conclusions, Challenges, and Future Research Orientations

### 5.1. Advantages, Limitations, and Drawbacks of Various Valorisation Techniques

Valorisation techniques can basically be divided into two types (conventional and novel-based). The technique preferred will depend on the parts of the fruit plants to be converted into useful bioactive compounds, mostly based on the moisture content and composition of the food waste to avoid energy-intensive processes [101,103]. For example, jackfruit leaves, barks, and fruits can be easily valorised using various conventional methods such as maceration, percolation, decoction, reflux, and through soxhlet extraction to develop various medicines for healthcare needs, while other bioactive compound extraction needs the employment of advanced technologies [101–104]. The sustainability of the valorisation process is of paramount importance for efficient reducing, reusing, and recycling, and in transitioning to a circular bioeconomy. The end-use products derived from the fruit waste and scale of production also need to be looked from the environmental, ecological, and social perspectives. The green technology/novel technologies employed are always more eco-friendly than the conventional approaches [95,98].

The application of the efficient bioconversion of jackfruit waste (such as peels) can generate varieties of useful material by facilitating the microbial fermentation process, and this material can be applied for reductions in water pollutants (such as dye/colour removal under bioremediation) in contaminated aquatic systems. This is a good example of a green economic model that is used for waste utilization [9,12]. A number of studies have claimed the utilization feasibility of jackfruit waste and this has been converted into value-added products via reducing/mitigating the waste quantity with the protection of the environment in a sustainable mode. Huge quantities of jackfruit waste are generated due to modern practices and now interests have developed in converting these into varieties of applications; jackfruit waste capabilities have a diverse array of functions [98]. As discussed in an earlier section, jackfruit waste also possesses good amounts of nutrients and varieties. The waste is a good source of carbohydrates, proteins, or minerals, and these nutrients are utilized for cultivating diverse groups of microbial systems/groups that can produce various metabolites such as organic acids, polysaccharides, enzymes, or therapeutic compounds [9,98]. In the conversion processes of jackfruit waste, different types of pre-treatments or bioprocesses such as physico-chemical, biological, and innovative green pre-treatment methods are discussed [97]. Further, this requires a deeper investigation of more genetically and biotechnologically advanced methods that can utilize the complete fractions of jackfruit waste and convert 100% of this waste into different products in direct ways with more functional natures. Further, more research is needed on advances in technology, equipment, and methodologies that can make the necessary changes with the promotion of biorefineries with the generation of varieties of value-added products [9,97].

### 5.2. Prospect of Jackfruit Waste as a Bio-Absorbent for Pollution Control

Recent studies have shown that jackfruit peel is an effective, low-cost bio-absorbent for the removal of chromium and nickel from an aqueous solution. Additionally, since jackfruit waste is a rich source of pectin, it can be useful for the economic removal of cadmium from water bodies/water subjected to various chemical pollutions [38]. Various reports

have claimed that jackfruit leaves (JLP) are an important waste and are suitable as an agro-waste categories-based material that can be utilized for the efficient removal of metal (such as lead—Pb –II form) from wastewater. Due to the surface medication of jackfruit leaf power (JLP), it can ably remove a high percentage of lead concentration [126]. Further, the surface modification of jackfruit leaves has been achieved through the application of chemical reagents such as isopropyl alcohol (20%), followed by treatment with alkali sodium hydroxide (AIJLP) and tartaric acid (TIJLP). These modifications were confirmed by the BET surface area (29 $m^2$/g). This chemical modification resulted in 1.5-fold and 2.5-fold increases in the surface area due to AIJLP (50 $m^2$/g) and TIJLP (72 $m^2$/g) impacts, respectively [127]. After generating low-cost bioabsorbents from JLP, these can be applied in batch experiments to confirm lead adsorption. There have been efforts too to find the optimal equilibrium conditions such as pH, contact time, and lead concentration. Similarly, from the developed bioabsorbsent, pore size and pore volume can also be analysed using Barrett–Joyner–Halenda (BJH) models/methods [126,127]. The BJH model has helped to provide the optimal parameters for the adsorption of toxic metal concentrations via this biomaterial property mechanism. Recently, in reusability studies, it was confirmed that AIJLP can efficiently remove lead contaminants/pollutants by up to 95% in up to five cycles. These results have recommended the application of AIJLP for bioremediation in wastewater [127]. The cost of developing bioadsorbents from jackfruit leaves and their subsequent modifications with chemical reagents can cost up to 11 USD/kg. This includes the cost of the collection of raw material, the washing and drying processes, and some miscellaneous expenses until the chemical reagents are produced in their final form. This is a very good example of low-cost bioadsorbents from jackfruit leaves [128,129]. Hence, the sustainable utilization of jackfruit waste biomass can contribute to environmental sustainability and energy access for rural populations [130]. The biowaste materials of jackfruit byproducts act as sustainable sources for various bioenergy and biochemical purposes [131–137].

### 5.3. Conclusions and Future Research Orientations

This paper mainly focused on jackfruit waste as a potential and sustainable source of augmenting energy requirements and meeting global challenges. There are plenty of jackfruit biowastes available in the world for the sustainable conversion of jackfruit waste into diverse bioactive compounds. Jackfruit trees can be found growing in wide ecological regions, and they can easily be raised as plantations with minimum agro-care. Nevertheless, many countries are aggressively increasing their jackfruit plantations to meet growing demand, thereby providing the availability of huge quantities of jackfruit waste. However, it becomes a big challenge to utilize these wastes from the environment. The long-term accumulation of these wastes in aquatic bodies can become a big threat to aquatic animals with the creation of water pollution. To avoid these issues, researchers have now applied many valorisation approaches to recover biofuels/other biochemicals in a sustainable manner. These efforts can help in the promotion of zero-waste generation from the complete utilization of jackfruit wastes with the production of bioactive compounds. These efforts toward value-added products from jackfruit waste can be found to be cheap sources of feedstock with reduced bioproduct prices. In the coming period, many green techniques such as MAE, UAE, and SCF-AE will be applied with green extraction solvents to obtain more effective bioproducts with high yields compared to traditional techniques. Different forms of jackfruit waste can be mitigated in a systematic manner via the synthesis of a number of value-added products, including cheap sources of adsorbents, nano-particles, and also biogases and other biofuels. The energy sources from jackfruit waste can be achieved via microbial fermentation with specific microbial cell systems and anaerobic digestion. Some applications of biochar have been discussed in relation to the soil amendment process, which increases soil fertility and also carbon/nitrogen content. This review also provides several ideas for sustainable fuel development with the maintenance of a clean environment. Furthermore, insights have been provided in this review into the different natures

of jackfruit waste sources, which have been analysed/studied for nutrient determination. Further, different valorisation techniques such as green extraction, the preparation of bioactive material approaches, and microbial fermentation were discussed in greater depth with mechanisms for generating/recovering nutrients/bioproducts. These conversion approaches have been found to be very effective in generating substantial amounts of bioproducts from these wastes, thus making the environment clean and safe. This is a good strategy for achieving zero waste in different conversion processes. The objectives of this review were achieved by a detailed focus on the mitigation of jackfruit waste/the recovery of nutrients with sustainable product development involving a zero-waste process.

**Author Contributions:** P.K.S., R.K.S. and A.K.S.: conceptualization; design of the study; formal data analysis; writing original draft, review, and editing. U.K.S.: data curation; formal analysis; methodology and typo—correction. P.P. and P.D.: methodology; review and editing of the manuscript. All authors have read and agreed to the published version of the manuscript.

**Funding:** This work has received no specific funding.

**Institutional Review Board Statement:** There is no institutional review board statement.

**Informed Consent Statement:** There is no informed consent statement.

**Data Availability Statement:** Not applicable.

**Acknowledgments:** The authors acknowledge the support received from their concerned universities for writing this report.

**Conflicts of Interest:** The authors declare no conflict of interest.

## Abbreviations

**AgNO$_3$:** silver nitrate; **AgNPs:** silver nanoparticles; **AHP:** alkaline hydrogen peroxide; **ASPE:** *Artocarpus heterophyllus* seed powder extract; **BMP:** biochemical methane potential; **BWR:** bread waste residues; **CF:** crude fibre; **CP:** critical point; **CP:** crude protein; **DPPH:** a,a-Diphenyl-β-picrylhydrazyl; **EAE:** enzyme-assisted extraction; **FTIR**: Fourier-transform infrared; **GA:** Genetic algorithm; JSP: jackfruit seed powder; **MAE:** microwave-assisted extraction; **MB:** methylene blue; **MCC:** microcrystalline cellulose; **NFE:** nitrogen-free extract; **Pc:** pressure point; **PEF-AE:** pulse electric field-assisted extraction; **PHA:** Polyhydroxyalkanoates; **POJ:** peel of jackfruit; **PSO:** Particle Swarm optimization; **RSM:** Response Surface Methodology; **SC-FE:** supercritical fluid extraction process; **SE:** Soxhlet extraction; **SEM**: scanning electron microscopy; **SHF:** separate fermentation hydrolysis; **Tc:** specific temperature; **UAE:** ultrasonic-assisted extraction; and **XRD:** X-ray diffraction.

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
