# Peer review of "The Utilization of Jackfruit (Artocarpus heterophyllus L.) Waste towards Sustainable Energy and Biochemicals: The Attainment of Zero-Waste Technologies"

_sustainability, doi:10.3390/su151612520_

Round 1

Reviewer 1 Report (Previous Reviewer 1)

While the valorization of food and fruit wastes, including jackfruit waste, is an important research area, the novelty of your study is limited. The concept of converting jackfruit waste into energy and biochemicals has been previously explored in the literature. Your manuscript does not sufficiently demonstrate novel insights or significant advancements in this field, which diminishes its contribution to the existing knowledge base.

Moderate editing of English language required

Author Response

Dear Reviewer,

Thank you very much for sparing your valuable time, and providing us numerous good suggestions. I wish to mention that we have revised the manuscript by incorporating all the suggestions. The changes made to the manuscript are highlighted in yellow color. Please find the details in the attached file.

Thank you, and with best regards,

--

Attached File: 

Author’s Response to the Esteemed Reviewer # 1

Manuscript Reference No: Sustainability-2503343 (Revised Manuscript)

Title of manuscript:  Utilization of Jackfruit (Artocarpus heterophyllus L.) waste to-wards sustainable energy and biochemicals: Attainment of zero waste technologies

Author’s reply

We are very much thankful to the reviewers for their valuable comments, which help us immensely to improve our manuscript. As per reviewer’s comments, we have now thoroughly revised our manuscript. All the comments have been looked upon and replied accordingly. The revisions are highlighted in yellow colour in the text of the revised manuscript.

REVIEWER COMMENTS # 1: Comments and Suggestions for Authors

Comment 1: While the valorization of food and fruit wastes, including jackfruit waste, is an important research area, the novelty of your study is limited. The concept of converting jackfruit waste into energy and biochemicals has been previously explored in the literature. Your manuscript does not sufficiently demonstrate novel insights or significant advancements in this field, which diminishes its contribution to the existing knowledge base.

Author’s response: We thank the esteemed reviewer very much for bringing this to our attention. The aforementioned suggestions have been incorporated in the introduction section of revised manuscript as per valuable comments.

Reviewer 2 Report (Previous Reviewer 2)

The authors corrected very much the manuscript. For moderate revision.

But it needs some improvements:

1. Introduction: Please use italics in the Latin names of organisms

2. Sticking words with values please correct

3. There is a lack of an author contribution statement.

The names of scientific databases should be from capital letters

Author Response

Dear Reviewer,

Thank you very much for sparing your valuable time, and providing us numerous good suggestions. I wish to mention that we have revised the manuscript by incorporating all the suggestions. The changes made to the manuscript are highlighted in yellow color. Please find the details in the attached file.

Thank you, and with best regards,

--

Attached File: 

Author’s Response to the Esteemed Reviewer # 2

Manuscript Reference No: Sustainability-2503343 (Revised Manuscript)

Title of manuscript:  Utilization of Jackfruit (Artocarpus heterophyllus L.) waste to-wards sustainable energy and biochemicals: Attainment of zero waste technologies

Author’s reply

We are very much thankful to the reviewers for their valuable comments, which help us immensely to improve our manuscript. As per reviewer’s comments, we have now thoroughly revised our manuscript. All the comments have been looked upon and replied accordingly. The revisions are highlighted in yellow colour in the text of the revised manuscript.

REVIEWER COMMENTS # 2: Comments and Suggestions for Authors

The authors corrected very much the manuscript. For moderate revision. But it needs some improvements:

Comment 1:  Introduction: Please use italics in the Latin names of organisms

Author’s response: We thank the esteemed reviewer very much for bringing this to our attention. The aforementioned suggestions have been incorporated throughout the revised manuscript as per valuable comments.

Comment 2:  Sticking words with values please correct

Author’s response: We thank the esteemed reviewer very much for bringing this to our attention. The manuscript has been revised thoroughly to resolve the aforementioned issues as per valuable comments.

Comment 3:  There is a lack of an author contribution statement.

Author’s response: We thank the esteemed reviewer very much for bringing this to our attention. The aforementioned suggestions have been incorporated after the conclusion section of the revised manuscript as per valuable comments.

Reviewer 3 Report (New Reviewer)

Sarangi et al discussed the utilization of jackfruit for various applications. The use of biomass feedstocks and subsequent utilization thereof is currently hot topic and therefore this article deserves recognition in this field of research. However, the reviewer advises an extensive major revision before the article can be published in Sustainability. As the article is full of grammatical and spelling errors, the article should be revised for spelling and grammar accordingly. The reviewer also suggests a proofread by a native English speaker. A second major remark is that the authors should carefully go through the text and place references alongside (quantitative) statements made.

-    -     The authors should go carefully through the introduction section and place references with statements made.

-    -     Something is not grammatically correct in Lines 56-59, Lines 59-60, Lines 121-123, Lines 185-186, 458-460, 462-463, 463-465 and not limited to these examples).

-   -      Line 54 (also mentioned later in the manuscript), is NFE not commonly referred to as Nitrogen Free Extract instead of Nitrogen Free Radical?

-   -      Line 84, Staphylococcus should be placed in italic

-   -      Line 86, test in plural

-  -       Line 130, it should be “the literature”

-  -       Line 176, “Areas” should not have a capital letter.

-   -      It would be interesting to update figure 1 and/or figure 2 with the % of biomas of each of the parts that could be derived from the jackfruit plant.

-   -      Line 245, “wastes” should not be in plural

-   -      Line 264 replace “Still time, …” with “Up until now, …”.

-  -       Figure 2; it would be more interesting to mention the amount of energy that can be produced per kg jackfruit shells via anaerobic digestion than providing 1 total number (this does not mean anything on its own).

-     -    Line 303 “others sugar” should be “other sugars”.

-   -      Typo Line 318 “licluding”.

-  -       Lines 322-324, present the same information.

-   -      Lines 322-329 should be rewritten as the same information is mentioned a few times. Please condense to 1 or 2 sentences.

-    -     What is meant with Lines 330-331?

-   -      Lines 336 - 344 are not really within the scope of this manuscript.

-   -      Are the %’s shown in Figure 3 based on dry matter or … ?

-   -      Line 369; the reviewer assumes it should be “ultrasound-assisted extraction” instead of “ultraviolet-assisted extraction” ?

-   -      Line 371 remove “,” after reference bracket.

-   -      Line 392, what is meant with the EIE process?

-  -       Line 395: typo? Should be “up to 87%” instead of “p to 87%”?

-   -      Line 399 replace “ultrasonic-assisted extraction” by “ultrasound-assisted extraction.”

-  -       Figure 4: replace ultraviolet-assisted extraction by ultrasound assisted extraction.

-  -       Line 452, should be “wine making process” instead of “wind making process”.

-  -       “MAE technique is an endothermic and spontaneous process and it is also influenced by operating conditions like temperature and feed ratio on flavonoid yield.” The reviewer does not understand what the author means by this statement. Applying microwaves in a MAE is a spontaneous process?

-  -       Typo Line 470 in °C.

-   -      Typos in Line 506 (bracket, point and capital letter).

-  -       Line 516, disrupt would be a better word compared to “hydrolyse” as not every pretreatment technology is focused on hydrolysis.

-  -       Line 543, “vaporization technology” or “valorization technology”?

- -        Line 547, the reviewer assumes that 0-80% non-edible parts is not correct? That a number is missing before “0”?

-  -       Line 651: Saccharomyces cerevisiae should be placed in italic.

- -        Line 667 to be consistent place Algorithm and Optimization with a capital letter.

- -        Line 678: Cow drug?? Should this be cow dung?

- -        Table 1, first row, should be “and so on” instead of “are so on”?

-   -      The authors state in Table 1 that biofuel is cheaper than fossil fuels. This is not (always) the case, and therefore the reviewer suggests to eliminate this statement from the text.

-  -       Something is wrong in row 2 of Table 1: 2.2 1010, is this correct?

-   -      Table 1, row 3: Bacillus subtilis should be placed in italic.

-   -      Table 1, row 4 use “-“ instead of “=” in bio-oil.

-  -       Spacing missing in Row 6, table 1.

-  -       Table 1, row 7, Artocarpus heterophyllus should be placed in italic. (likewise in the next rows).

-  -       Line 758 spacing missing

-  -       Table 2, place Artocarpus heterophyllus everywhere in italic.

-  -       Line 848: “And this model has helped the adsorption mechanism.” How does a model help the adsorption mechanism?

See comments above.

Author Response

Dear Reviewer,

Thank you very much for sparing your valuable time, and providing us numerous good suggestions. I wish to mention that we have revised the manuscript by incorporating all the suggestions. The changes made to the manuscript are highlighted in yellow color. Please find the details in the attached file.

Thank you, and with best regards,

--

Attached File: 

Author’s Response to the Esteemed Reviewer # 3

Manuscript Reference No: Sustainability-2503343 (Revised Manuscript)

Title of manuscript:  Utilization of Jackfruit (Artocarpus heterophyllus L.) waste to-wards sustainable energy and biochemicals: Attainment of zero waste technologies

Author’s reply

We are very much thankful to the reviewers for their valuable comments, which help us immensely to improve our manuscript. As per the reviewer’s comments, we have now thoroughly revised our manuscript. All the comments have been looked upon and replied accordingly. The revisions are highlighted in yellow colour in the text of the revised manuscript.

REVIEWER COMMENTS # 3: Comments and Suggestions for Authors

Sarangi et al discussed the utilization of jackfruit for various applications. The use of biomass feedstocks and subsequent utilization thereof is currently hot topic and therefore this article deserves recognition in this field of research. However, the reviewer advises an extensive major revision before the article can be published in Sustainability. As the article is full of grammatical and spelling errors, the article should be revised for spelling and grammar accordingly. The reviewer also suggests a proofread by a native English speaker. A second major remark is that the authors should carefully go through the text and place references alongside (quantitative) statements made.

Comment 1:  The authors should go carefully through the introduction section and place references with statements made.

Author’s response: We thank the esteemed reviewer very much for bringing this to our attention. The introduction section of the present manuscript has been now revised as per valuable comments.

Comment 2:  Something is not grammatically correct in Lines 56-59, Lines 59-60, Lines 121-123, Lines 185-186, 458-460, 462-463, 463-465 and not limited to these examples).

Author’s response: We thank the esteemed reviewer very much for bringing this to our attention. The manuscript has been now revised as per valuable comments.

Comment 3:  Line 54 (also mentioned later in the manuscript), is NFE not commonly referred to as Nitrogen Free Extract instead of Nitrogen Free Radical?

Author’s response: We thank the esteemed reviewer very much for bringing this to our attention. The manuscript has now been corrected as per valuable comments.

Comment 4: Line 84, Staphylococcus should be placed in italic.

Author’s response: We thank the esteemed reviewer very much for bringing this to our attention. The correction has been made in the manuscript as per valuable comments.

Comment 5: Line 86, test in plural.  

Author’s response: We thank the esteemed reviewer very much for bringing this to our attention. The correction has been made in the manuscript as per valuable comments.

Comment 6: Line 130, it should be “the literature”.

Author’s response: We thank the esteemed reviewer very much for bringing this to our attention. The manuscript has now been corrected as per valuable comment.

Comment 7:  Line 176, “Areas” should not have a capital letter.

Author’s response: We thank the esteemed reviewer very much for bringing this to our attention. The correction has been made in the manuscript as per valuable comments.

Comment 8: It would be interesting to update figure 1 and/or figure 2 with the % of biomass of each of the parts that could be derived from the jackfruit plant.

Author’s response: We thank the esteemed reviewer very much for the same suggestion. Based upon the available information from the scientific literature, the same figures in the manuscript have been now modified/ revised as per valuable comments.

Comment 9: Line 245, “wastes” should not be in plural.

Author’s response: We thank the esteemed reviewer very much for bringing this to our attention. The correction has been made in the manuscript as per valuable comments.

Comment 10: Line 264 replace “Still time, …” with “Up until now”  

Author’s response: We thank the esteemed reviewer very much for the same suggestion. The correction has been made in the manuscript as per valuable comments.

Comment 11: Figure 2; it would be more interesting to mention the amount of energy that can be produced per kg jackfruit shells via anaerobic digestion than providing 1 total number (this does not mean anything on its own).

Author’s response: We thank the esteemed reviewer very much for the same suggestion. Based upon the available information from the scientific literature, the same figure of the manuscript has been now modified/ revised as per valuable comments.

Comment 12: Line 303 “others sugar” should be “other sugars”.  

Author’s response: We thank the esteemed reviewer very much for the same suggestion. The correction has been made in the manuscript as per valuable comments.

Comment 13:  Typo Line 318 “licluding”.

Author’s response: We thank the esteemed reviewer very much for bringing this to our attention. The correction has been made in the manuscript as per valuable comments.

Comment 14: Lines 322-324, present the same information.

Author’s response: We thank the esteemed reviewer very much for bringing this to our attention.  Sentences with the same information have been removed in the revised manuscript as per valuable comment.

Comment 15: Lines 322-329 should be rewritten as the same information is mentioned a few times. Please condense to 1 or 2 sentences.  

Author’s response: We thank the esteemed reviewer very much for bringing this to our attention.  Sentences have been condensed to 1 or 2 sentences in the revised manuscript as per valuable comments.

Comment 16:  What is meant with Lines 330-331?

Author’s response: We thank the esteemed reviewer very much for bringing this to our attention. Now it is modified for a clear understanding in the revised manuscript as per valuable comments.

Comment 17: Lines 336 - 344 are not really within the scope of this manuscript

Author’s response: We thank the esteemed reviewer very much for bringing this to our attention. The same text portions have been deleted as per valuable comments.

Comment 18: Are the %’s shown in Figure 3 based on dry matter or … ?  

Author’s response: We thank the esteemed reviewer very much for bringing this to our attention. It is not based on dry matter %. It is just % of edible and non-edible portions in plants/ plant fruit and accordingly revised in the manuscript

Comment 19:  Line 369; the reviewer assumes it should be “ultrasound-assisted extraction” instead of “ultraviolet-assisted extraction” ?

Author’s response: We thank the esteemed reviewer very much for bringing this to our attention. Theultraviolet-assisted extraction” has been replaced with “ultrasound-assisted extraction” in the revised manuscript as per valuable comments.

Comment 20:  Line 371 remove “,” after reference bracket.

Author’s response: We thank the esteemed reviewer very much for bringing this to our attention. The correction has been made in the revised manuscript as per valuable comments.

Comment 21: Line 392, what is meant with the EIE process?

Author’s response: We thank the esteemed reviewer very much for bringing this to our attention. The correction has been made in the revised manuscript as per valuable comments. In this context, “EIE” is not a correct abbreviation, and therefore corrected to EAE (enzyme-assisted extraction) in the revised manuscript.

Comment 22:  Line 395: typo? Should be “up to 87%” instead of “p to 87%”?

Author’s response: We thank the esteemed reviewer very much for bringing this to our attention. The correction has been made to “87%” instead of “up to 87%” in the revised manuscript as per valuable comments.

Comment 23: Line 399 replace “ultrasonic-assisted extraction” by “ultrasound-assisted extraction.”  

Author’s response: We thank the esteemed reviewer very much for bringing this to our attention. The correction has been made in the revised manuscript as per valuable comments.

Comment 24: Figure 4: replace ultraviolet-assisted extraction by ultrasound assisted extraction.

Author’s response: We thank the esteemed reviewer very much for bringing this to our attention. The correction has been made in the same figure of the revised manuscript as per valuable comments.

Comment 25: Line 452, should be “wine making process” instead of “wind making process”.

Author’s response: We thank the esteemed reviewer very much for bringing this to our attention. The correction has been made in the revised manuscript as per valuable comments.

Comment 26: “MAE technique is an endothermic and spontaneous process and it is also influenced by operating conditions like temperature and feed ratio on flavonoid yield.” The reviewer does not understand what the author means by this statement. Applying microwaves in a MAE is a spontaneous process?

Author’s response: We thank the esteemed reviewer very much for bringing this to our attention. The correction has been made in the revised manuscript as per valuable comments.

Comment 27: Typo Line 470 in °C.

Author’s response: We thank the esteemed reviewer very much for bringing this to our attention. The correction has been made in the revised manuscript as per valuable comments.

Comment 28:  Typos in Line 506 (bracket, point and capital letter).

Author’s response: We thank the esteemed reviewer very much for bringing this to our attention. The correction has been made in the revised manuscript as per valuable comments.

Comment 29: Line 516, disrupt would be a better word compared to “hydrolyse” as not every pretreatment technology is focused on hydrolysis.

Author’s response: We thank the esteemed reviewer very much for bringing this to our attention. The correction has been made to “disrupt” in the revised manuscript as per valuable comments.

Comment 30: Line 543, “vaporization technology” or “valorization technology”?

Author’s response: We thank the esteemed reviewer very much for bringing this to our attention. The correction has been made to “valorization technology” in the revised manuscript as per valuable comments.

Comment 31:  Line 547, the reviewer assumes that 0-80% non-edible parts is not correct? That a number is missing before “0”?

Author’s response: We thank the esteemed reviewer very much for bringing this to our attention. The correction has been made to “70-80%” in the revised manuscript as per valuable comments.

Comment 32: Line 651: Saccharomyces cerevisiae should be placed in italic.

Author’s response: We thank the esteemed reviewer very much for bringing this to our attention. The correction has been made in the revised manuscript as per valuable comments.

Comment 33: Line 667 to be consistent place Algorithm and Optimization with a capital letter.

Author’s response: We thank the esteemed reviewer very much for bringing this to our attention. The correction has been made in the revised manuscript as per valuable comments.

Comment 34: Line 678: Cow drug?? Should this be cow dung?  

Author’s response: We thank the esteemed reviewer very much for bringing this to our attention. The correction has been made to “cow dung” in the revised manuscript as per valuable comments.

Comment 35: Table 1, first row, should be “and so on” instead of “are so on”?

Author’s response: We thank the esteemed reviewer very much for bringing this to our attention. The correction has been made to “and so on” in the revised manuscript as per valuable comments.

Comment 36: Something is wrong in row 2 of Table 1: 2.2 1010, is this correct?

Author’s response: We thank the esteemed reviewer very much for bringing this to our attention. The correction has been made to “2.2 x 1010” in the revised manuscript as per valuable comments.

Comment 37: Table 1, row 3: Bacillus subtilis should be placed in italic.

Author’s response: We thank the esteemed reviewer very much for bringing this to our attention. The correction has been made in the revised manuscript as per valuable comments.

Comment 38: Table 1, row 4 use “-“ instead of “=” in bio-oil.

Author’s response: We thank the esteemed reviewer very much for bringing this to our attention. The correction has been made to “bio-oil” in Table 1of the revised manuscript as per valuable comments.

Comment 39:  Spacing missing in Row 6, table 1.

Author’s response: We thank the esteemed reviewer very much for bringing this to our attention. The correction has been incorporated in Table 1 of the revised manuscript as per valuable comments.

Comment 40: Table 1, row 7, Artocarpus heterophyllus should be placed in italic. (likewise in the next rows).  

Author’s response: We thank the esteemed reviewer very much for bringing this to our attention. The correction has been incorporated in Table 1 of the revised manuscript as per valuable comments.

Comment 41: Line 758 spacing missing

Author’s response: We thank the esteemed reviewer very much for bringing this to our attention. The correction has been incorporated in the revised manuscript as per valuable comments.

Comment 42: Table 2, place Artocarpus heterophyllus everywhere in italic.  

Author’s response: We thank the esteemed reviewer very much for bringing this to our attention. The suggested correction has been incorporated in the revised manuscript as per valuable comments.

Comment 43:  Line 848: “And this model has helped the adsorption mechanism.” How does a model help the adsorption mechanism?

Comment 44: We thank the esteemed reviewer very much for bringing this to our attention. The Barrett-Joyner-Halenda (BJH) model has helped to provide the optimal parameters for the adsorption of toxic metals concentration via the biomaterial properties mechanism. This correction has been incorporated in the revised manuscript as per valuable comments.

Reviewer 4 Report (New Reviewer)

Review Comments: 

The objective of the manuscript entitled: “Utilization of Jackfruit (Artocarpus heterophyllus L.) waste towards sustainable energy and biochemicals: Attainment of zero  waste technologies”

The research was well performed, and the subject is interesting and up to date, however, English is very difficult to understand and requires firstly a major language revision, after which the manuscript can be taken into account for further revision.

The research was well performed, and the subject is interesting and up to date, however, English is very difficult to understand and requires firstly a major language revision, after which the manuscript can be taken into account for further revision and possible publication.

Author Response

Dear Reviewer,

Thank you very much for sparing your valuable time, and providing us numerous good suggestions. I wish to mention that we have revised the manuscript by incorporating all the suggestions. The changes made to the manuscript are highlighted in yellow color. Please find the details in the attached file.

Thank you, and with best regards,

--

Attached File: 

Author’s Response to the Esteemed Reviewer # 4

Manuscript Reference No: Sustainability-2503343 (Revised Manuscript)

Title of manuscript: Utilization of Jackfruit (Artocarpus heterophyllus L.) waste to-wards sustainable energy and biochemicals: Attainment of zero waste technologies

Author’s reply

We are very much thankful to the reviewers for their valuable comments, which help us immensely to improve our manuscript. As per reviewer comments, we have now thoroughly revised our manuscript. All the comments have been looked upon and replied accordingly. The revisions are highlighted in yellow colour in the text of the revised manuscript.

REVIEWER COMMENTS # 4: Comments and Suggestions for Authors

Comment 1: The objective of the manuscript entitled: “Utilization of Jackfruit (Artocarpus heterophyllus L.) waste towards sustainable energy and biochemicals: Attainment of zero waste technologies”

The research was well performed, and the subject is interesting and up to date, however, English is very difficult to understand and requires firstly a major language revision, after which the manuscript can be taken into account for further revision.

Author’s response: We thank the esteemed reviewer very much for bringing this to our attention. We wish to inform that we have revised the language part of the manuscript.

Round 2

Reviewer 3 Report (New Reviewer)

The authors have addressed most of the reviewer's comments and therefore the reviewer suggests to accept this manuscript for publication. However, it should be noted that there are still some spelling and grammar mistakes throughout the text.

See comment above.

Author Response

Dear Reviewer,

Thank you very much for recommending acceptance of our manuscript. We are nevertheless grateful to you.

As per your advice, we have thoroughly checked the manuscript for grammatical errors, and spelling mistakes, and have now rectified them in the revised manuscript.

We thank you once again.

With best regards,

Professor U.K.Sahoo

Reviewer 4 Report (New Reviewer)

Accept in present form. 

Author Response

Dear Reviewer,

We thank you immensely for recommending acceptance of our manuscript in its present form.

With best regards,

Professor U.K.Sahoo

This manuscript is a resubmission of an earlier submission. The following is a list of the peer review reports and author responses from that submission.

Round 1

Reviewer 1 Report

After careful evaluation, I regret to inform that paper is not suitable for publication. While I appreciate your efforts in highlighting the potential valorization of jackfruit waste, we have identified several concerns and limitations that need to be addressed before considering it for publication.

Specifically, the reasons for rejection are as follows:

  1. Lack of Novelty: The paper does not demonstrate sufficient novelty in terms of the valorization of jackfruit waste. Although the potential conversion of fruit waste into sustainable energy and biochemicals is an important topic, the approach and findings presented in this paper do not sufficiently differentiate themselves from existing studies and fail to provide significant new insights or advancements in the field.

  2. Lack of Comparative Analysis: The paper does not sufficiently compare the proposed conversion technologies for jackfruit waste with existing alternatives. A comprehensive analysis of the advantages, limitations, and potential drawbacks of the proposed technologies is necessary to evaluate their viability and potential for practical implementation.

  3. Limited Discussion on Environmental Implications: Although the paper briefly mentions the contribution of jackfruit waste to disposal and pollution issues, there is a lack of in-depth analysis and discussion on the environmental implications of the proposed valorization methods. Considering the importance of sustainable solutions, it is crucial to address the environmental impacts and potential mitigation strategies associated with the proposed technologies.

  4. Inadequate Conclusion: The conclusion of the paper does not provide a comprehensive summary of the main findings and their implications. It is essential to clearly articulate the significance and potential impact of the research, as well as suggest avenues for future research and practical applications.

Reviewer 2 Report

Please improve due to points:

1. Please add the Latin name of the jackfruit tree for full understanding in the title or abstract.

2. Please add the methodology part where you add methods of selection of chosen references.

3. Please add more aims of research in the introduction

Please use superscript for C grades, use numerical values 80-90% instead of eighty and ninety.

Reviewer 3 Report

1. You've explained how the carbon dioxide produced during the pyrolysis process of Bread Waste Residues (BWR) reacts with the volatile matter evolved from BWR, which in turn reduces the concentration of CO2. Could you provide more details on the specific mechanisms that allow the reduction of CO2 during this stage of the process? Understanding the detailed mechanism may provide deeper insights into how this process can be optimized for better results.

2. You mention the production of various biofuels through microbial fermentation. Could you please elaborate on the types of microbes that are effective in this process? Are there any specific microbes that are particularly efficient in fermenting jackfruit waste into biofuel? If yes, what might be the reasons for their efficiency?

3. The description of the pretreatment process, particularly the changes in matter after the addition of chemical solvents, was particularly insightful. However, there's an aspect that I think requires further explanation: Could you provide more details or examples on the types of bioproducts (apart from the ones thas have been mentioned) that can be generated from jackfruit waste?